# Distributed Estimation with Multiple Samples per User: Sharp Rates and Phase Transition

**Jayadev Acharya**
Cornell University
acharya@cornell.edu

**Clément L. Canonne**
University of Sydney
clement.canonne@sydney.edu.au

**Yuhan Liu**
Cornell University
yl2976@cornell.edu

**Ziteng Sun**
Cornell University
zs335@cornell.edu

**Himanshu Tyagi**
Indian Institute of Science
htyagi@iisc.ac.in

## Abstract

We obtain tight minimax rates for the problem of distributed estimation of discrete distributions under communication constraints, where $n$ users observing $m$ samples each can broadcast only $\ell$ bits. Our main result is a tight characterization (up to logarithmic factors) of the error rate as a function of $m$, $\ell$, the domain size, and the number of users under most regimes of interest. While previous work focused on the setting where each user only holds one sample, we show that as $m$ grows the $\ell_1$ error rate gets reduced by a factor of $\sqrt{m}$ for small $m$. However, for large $m$ we observe an interesting phase transition: the dependence of the error rate on the communication constraint $\ell$ changes from $1/\sqrt{2^\ell}$ to $1/\sqrt{\ell}$.

## 1  Introduction

### 1.1  Background and motivation

We consider the problem of *density estimation* in a distributed setting where each of $n$ users locally holds a dataset of $m$ independent observations from an unknown discrete probability distribution, and a central server seeks to leverage their data in order to estimate this underlying distribution. This abstract formulation encompasses an array of crucial building blocks in areas such as federated learning, IoT, or measurement from deployed sensors, and has (along with some of its variants) received significant attention in recent years.

Our focus in this work is to understand how a *bandwidth constraint* at each user affects this learning task. Namely, to characterize the fundamental tradeoffs between *communication*, *local sample size*, and *estimation error rate*. Previous work on distributed estimation has focused primarily on the single-sample case where each user has exactly one draw from an unknown underlying distribution $\mathbf{p}$. We consider the more general, more practical, and technically more involved setting where each of the $n$ users holds $m \geq 1$ samples from a $k$-ary distribution, and can transmit $\ell$ bits to a central server. Our goal is to understand the minimax rates of estimating $\mathbf{p}$ in $\ell_1$ distance. Before detailing our results and discussing their relation with prior work, we provide the necessary background and notation.

### 1.2  Problem setting and notation

Let $k$ be a positive integer. Let $\Delta_k = \{\mathbf{p} = (\mathbf{p}_1, \ldots, \mathbf{p}_k) \in \mathbb{R}^k : \mathbf{p} \geq 0, \|\mathbf{p}\|_1 = 1\}$ denote the set of all probability distributions over a discrete domain of size $k$, which we assume without loss of generality is $[k] := \{1, 2, \ldots, k\}$. For $\mathbf{p}, \mathbf{q} \in \Delta_k$, the total variation (TV) distance between

35th Conference on Neural Information Processing Systems (NeurIPS 2021).

$\mathbf{p}$ and $\mathbf{q}$ is $\mathrm{TV}(\mathbf{p}, \mathbf{q}) := \sup_{S \subseteq [k]} (\mathbf{p}(S) - \mathbf{q}(S))$. Here for a set $S \subseteq [k]$, $\mathbf{p}(S) := \sum_{x \in S} \mathbf{p}_x$. Each of $n$ users observes $m$ i.i.d. samples from the same (unknown) distribution $\mathbf{p} \in \Delta_k$. We write $X_i = (X_i^{(1)}, X_i^{(2)}, \ldots, X_i^{(m)})$ for the samples of the $i$th user, and $X^n = (X_1, X_2, \ldots, X_n) \in [k]^{nm}$ for the collection of all $nm$ samples. We denote the empirical frequencies of samples at user $i$ by $M_i$, where, for $x \in [k]$, $M_i(x) := \frac{1}{m} \sum_{j=1}^{m} \mathbb{1}\{X_i^{(j)} = x\}$.

Each user can only send an $\ell$-bit message, capturing their bandwidth constraints. In particular, user $i$ is allowed to send a message $Y_i \in \{0,1\}^\ell$ to the central server through a *channel* $W_i \in \mathcal{W}_\ell$, that is, a Markov kernel $W_i \colon \{0,1\}^\ell \times [k]^m \to [0,1]$, where

$$W_i(y \mid x) = \Pr(Y_i = y \mid X_i = \mathbf{x}), \qquad \forall \mathbf{x} \in [k]^m, y \in \{0,1\}^\ell.$$

Intuitively $W_i$ can be viewed as a randomized function that maps an input $X_i$ to some $\ell$-bit string $Y_i$, which represents the messaging scheme used by the user. We consider two standard types of protocols, noninteractive and interactive. In *noninteractive* protocols, all the channels $W_i$s are chosen in parallel, and the messages are sent independently of each other with no shared random seed available. In *(sequentially) interactive* protocols, the choice of $W_i$ may depend on the previous messages $Y^{i-1} := (Y_1, Y_2, \ldots, Y_{i-1})$; that is, the message sent by the $i$th user can depend on the messages sent by the previous users, and on a common random seed $U$ (independent of the observations) available to all users.[1][2]

**Distribution estimation.** Upon observing the messages $Y^n := (Y_1, Y_2, \ldots, Y_n)$, the server estimates the underlying distribution $\mathbf{p}$ with an estimator $\widehat{\mathbf{p}} \colon (\{0,1\}^\ell)^n \to \Delta_k$, with goal to minimize the expected total variation (TV) distance between $\mathbf{p}$ and $\widehat{\mathbf{p}}$. In this work, we are interested in studying the *minimax risk* for distribution estimation under $\ell$-bit communication constraints, defined as

$$\mathcal{R}(\ell, k, n, m) := \min_{W^n \in \mathcal{W}_\ell^n} \min_{\widehat{\mathbf{p}}} \max_{\mathbf{p} \in \Delta_k} \mathbb{E}[\mathrm{TV}(\widehat{\mathbf{p}}(Y^n), \mathbf{p})] \tag{1}$$

where the minimum is taken over all (possibly interactive) protocols.

## 1.3 Prior results and our contribution

For the setting when $m = 1$, *i.e.*, each user only observes *one* sample, the problem has been studied by various works in the literature [11, 1, 3], where it has been established that

$$\mathcal{R}(\ell, k, n, 1) = \Theta\left(\sqrt{\frac{k}{n}} \vee \sqrt{\frac{k^2}{n2^\ell}}\right). \tag{2}$$

In this work, we consider a natural setting where each user has access to $m > 1$ i.i.d. samples. As a baseline, it is immediate to see that, absent any communication constraint (*i.e.*, $\ell = \infty$), the error rate would behave as in the centralized case,

$$\mathcal{R}(\infty, k, n, m) = \Theta\left(\sqrt{\frac{k}{mn}}\right). \tag{3}$$

This raises the natural question of how the minimax risk behaves between those two extremes, and of whether the exponential decrease of the risk as a function of the communication constraint $\ell$ is preserved. We point out that this setting is substantially more challenging to analyze, and requires significantly different techniques and new ideas than the $m = 1$ case. Indeed, as $m$ grows, even the case $\ell = 1$ is not entirely clear: given $1 \ll m \ll k$ samples and only one bit of information to send, should each user send that bit of information about one of their samples only? Or about the (quite imprecise) empirical distribution they can form with their own $m$ samples? Now, what about the case $m \gg k$, where each user now can compute by themselves an accurate estimate of the unknown $\mathbf{p}$, but still lack the bandwith to send enough information about it to the server?

---

[1]Formally, this is captured by defining channels as $W_i \colon \mathcal{Y} \times [k]^m \times \mathcal{Y}^{i-1} \times \mathcal{U} \to [0,1]$, where $\mathcal{Y} := \{0,1\}^\ell$.

[2]We note that some work further distinguish between *private-coin* noninteractive protocols, where each user randomizes their message independently as described above, and *public-coin* noninteractive protocols, where a common random seed is available to all users. In this work, we design private-coin noninteractive protocols for our upper bounds; as our lower bounds hold even for sequentially interactive protocols, this only makes our results stronger.

We are able to obtain tight bounds for most parameter regimes, and show that the minimax rates exhibit an interesting *phase transition* as $m$ and $\ell$ vary (with respect to the domain size $k$). We state the results under cases where $m < k/2^\ell$ and $m \geq k/2^\ell$ in Theorem 1.1 and 1.2 respectively.

**Theorem 1.1.** *When $m < k/2^\ell$ and $n/\log(n) > \frac{k}{2^\ell}\log m$, the minimax rate satisfies*

$$\mathcal{R}(\ell, k, n, m) = \Theta\left(\sqrt{\frac{k}{mn}} \vee \sqrt{\frac{k^2}{mn2^\ell}}\right).$$

*Moreover, this bound is achieved by a noninteractive communication protocol.*

We observe that, keeping the same communication budget for each user, the minimax risk is reduced by a factor of $\sqrt{m}$ compared to the one-sample case (2). In this regime, increasing the communication budget $\ell$ still reduces the estimation error exponentially, as it did in the one-sample case. Our second set of results shows that this is no longer the case as $m$ increases:

**Theorem 1.2.** *When $m \geq k/2^\ell$ and $n\ell/\log(n) > \min(m\log(k/m+1), k)\log m$, the minimax rate satisfies*

$$\mathcal{R}(\ell, k, n, m) = \begin{cases} O\left(\sqrt{\frac{k}{mn}} \vee \sqrt{\frac{k\log(k/m+1)}{n\ell}}\right), & \text{when } m < k, \\ O\left(\sqrt{\frac{k}{mn}} \vee \sqrt{\frac{k^2}{mn\ell}}\right), & \text{when } m \geq k. \end{cases}$$

*Moreover, this bound is achieved by a noninteractive communication protocol.*

Unlike the previous case $m < k/2^\ell$, the risk now decreases at a rate $1/\sqrt{\ell}$ instead of $1/\sqrt{2^\ell}$, an *exponential* slowdown. As stated, however, Theorem 1.2 only provides an upper bound on the minimax rate, from which it would still be conceivable that the true rate behaves as $1/\sqrt{2^\ell}$. We rule out this possibility, confirming the phase transition suggested by Theorem 1.2 as $m$ grows:

**Theorem 1.3.** *When $m \geq k/2^\ell$ and $n > (k/\ell)^2$, the minimax rate satisfies*

$$\mathcal{R}(\ell, k, n, m) = \begin{cases} \Omega\left(\sqrt{\frac{k}{mn}} \vee \sqrt{\frac{k}{n\ell\log k}}\right), & \text{when } m < k\log k, \\ \Omega\left(\sqrt{\frac{k}{mn}} \vee \sqrt{\frac{k^2}{mn\ell}}\right), & \text{when } m \geq k\log k. \end{cases}$$

Note that the lower bound of Theorem 1.3 matches the upper bound from Theorem 1.2 up to either constant or logarithmic factors. Interestingly, our results imply that, keeping other parameters fixed, increasing $m$ from $k/2^\ell$ to $k$ can at best improve the rate by logarithmic factors in $k$.

**About the restriction on $n$.** Our bounds for rates are tight up to logarithmic factors in $k$, for sufficiently large $n$. The constraint on the range of $n$ in Theorems 1.1 to 1.3 may appear restrictive at first. However, note that since Theorem 1.1 focuses on the regime $m < k/2^\ell$, the bound becomes vacuous for $n \leq k$ (as then $\frac{k^2}{mn2^\ell} \geq 1$), and thus we can always assume $n \geq k$. Thus, the requirement $n/\log n \geq \frac{k}{2^\ell}\log m$ is only restrictive for $\ell \ll \log\log m + \log\log n$. The assumption in Theorem 1.2 is more consequential; but it can be shown that $n \cdot \ell$ must be at least linear in $k$ under some conditions. Indeed, even if each user were given the distribution up to $\ell_1$ error $\varepsilon < 1$, a standard packing argument would require $n\ell \geq k\log\frac{1}{\varepsilon}$ bits of communication. Finally, the requirement $n > (k/\ell)^2$ (instead of the natural $n > k/\ell$ we just argued) in Theorem 1.3 is indeed not innocuous, and is an artifact of our proof method. For $m = k\log k$, note that if we want to estimate the distribution up to TV error $\varepsilon < \sqrt{\ell/(k\log k)}$, the above lower bound implies $n = \Omega\big(k/(\varepsilon^2\ell\log k)\big) > (k/\ell)^2$. Hence Theorem 1.3 demonstrates a phase transition in this case. When $\varepsilon > \sqrt{\ell/(k\log k)}$, the phase transition is less clear given Theorem 1.3 alone. However, observe that by the packing argument discussed above, we need at least $n \geq k\log\frac{1}{\varepsilon}/\ell$ users even when each user knows the exact distribution, and hence for any fixed $m$. This shows that the rate $O(\sqrt{k^2/(mn2^\ell)})$ with exponential dependence on $\ell$ cannot always hold. We see removing these restrictions as an interesting future direction.

**Our algorithm.** To motivate our algorithm, we consider the simplest distribution estimation problem: under communication constraints "For $nm$ coin tosses with each user observing $m$ coin tosses and

sending 1 bit to the center, how do we best estimate the bias of the coin?" A key component of our general algorithm is an algorithm for this simplest of settings, which turns out to be surprisingly non-trivial.

One may easily come up with simple solutions. One idea is to let each user send 0 if there are more heads than tails and 1 otherwise. This algorithm works well when the coin is almost fair, *e.g.*, $\Pr(\text{head}) \in (1/2 - 1/\sqrt{m}, 1/2 + 1/\sqrt{m})$, but it is ineffective if the coin is strongly biased. Another idea might be to only let each user indicate whether head appeared or not. This would lead to much better performance for strongly biased coins, *e.g.*, $\Pr(\text{head}) < 1/m$, but worse accuracy for fair coins. We elaborate further on this in Section 2.1.1.

In Section 2.1, we propose an algorithm which estimates the bias of the coin with accuracy $O(\sqrt{1/mn})$ with 1-bit of communication from each user under mild regularity conditions. Note that this is the best rate that can be achieve even when the server has access to all clean samples. Our proposed algorithm not only combines the best of the two naïve algorithms, but further tackles the regime where both solutions fail. It first forms local estimates of the bias of lower accuracy, and then boosts the accuracy by combining local estimates. For the first part, we use a scheme similar to the Gray coding based scheme proposed for Gaussian mean estimation in [8], but we need to modify it significantly as the tail probabilities of the underlying distribution can change with the unknown bias.

Using this algorithm as a primitive, we form a general algorithm in steps. The first step is to convert the bias estimation algorithm to that for estimating a $t$-ary distribution using 1-bit communication per user. We do so by partitioning the overall domain $[k]$ into $t \approx k/2^\ell$ parts of size at most $2^\ell$: we then estimate the distribution induced on the parts, using the 1-bit algorithm above by setting $t$ to equal the number of parts. (Users assigned to this task use their $m$ samples, but only one bit of communication.) That being done, we estimate the $k/2^\ell$ conditional distributions given the sample lies in each of these parts. Since each part has size at most $2^\ell$, the user can communicate one exact value in these parts using $\ell$ bits, which allows us to perform this task under our $\ell$-bit communication constraint, keeping only one of the $m$ samples and transmitting its exact value within a given part. (Users assigned to this task use their $\ell$ bits of communication, but only one of their $m$ samples.) Finally, combining the two types of estimates – of the induced distribution over parts, and of the conditional distribution within each part – allows us to form our overall estimate of the distribution.

## 1.4 Related Works

We first discuss the literature most relevant to our work. As previously mentioned, in the one-sample case ($m = 1$) discrete distribution estimation under communication constraints has received significant attention over recent years: [11], and [2, 3] obtain the tight minimax rates stated in Eq. (2), for the noninteractive case. [12] and [1] extend the lower bounds to interactive protocols (resp., in the blackboard model and the sequential interactivity model considered here), showing that the minimax error rate does not improve with interactivity. [4] builds upon [1] and extend this lower bound framework to more general distribution families; our own lower bounds will draw upon the techniques in this paper.

Switching from total variation (*i.e.*, $\ell_1$) loss to the less stringent $\ell_2$ loss, [6] develops a lower bound technique based on Fisher information, which allows us to pinpoint the minimax rates (again, in the $m = 1$ case) under $\ell_2$ loss for various distribution families, including discrete distributions. Finally, [10] states tight minimax bounds for estimation for $\ell_1$ loss under a different communication constraint, where each user can hold an unlimited number of samples but the constraint is on the *total* communication budget (as opposed to our setting, where each user has its own bandwidth constraint). We once again emphasize that all the techniques from the aforementioned papers are either limited to the $m = 1$ case or focusing on a different loss than $\ell_1$, and do not extend to the more challenging $m > 1$ case (or would lead to vacuous bounds due to the loose connection between $\ell_2$ and $\ell_1$ losses).

**Other works on communication-constrained estimation.** Beyond the case of discrete distribution estimation, there is a significant body of work on various parametric estimation tasks under communication constraints (including the case $m \geq 1$). [14] considers parameter estimation for the $\ell_2$ loss under a total communication budget constraint. In the same setting, [7] focuses on Gaussian mean estimation (note that for Gaussian distributions, having a number of samples $m > 1$ per user is less of a technical challenge, as the sum of $m$ samples remains Gaussian, with a different variance). [13] obtains results in the same communication model as ours (per-user communication constraint), but for high-dimensional product Bernoulli distributions. Finally, [8] studies parameter estimation for

high-dimensional Gaussian distributions under per-user communication constraints; some of their techniques, such as the use of a Gray code, are similar to ours; however, the product structure and stability by convolution of Gaussian distributions make the Gaussian setting fundamentally incomparable to the univariate discrete case.

## 2 Start with one bit: from multinomial to binomial

In this section, we consider the setting where each user can only send one bit. In particular, we present a one-bit algorithm which achieves the guarantee stated below; this algorithm will later be a key subroutine for the general case $\ell > 1$.

**Theorem 2.1.** *When $n > 100k \log(m) \log(n)$ and $\ell = 1$, the minimax rate satisfies*

$$\mathcal{R}(1, k, n, m) = O\left(\sqrt{\frac{k^2}{mn}}\right).$$

As described in the setting, each user observes $m$ samples from a multinomial distribution $\mathbf{p}$. Our algorithm then estimates $\mathbf{p}$ by learning each $\mathbf{p}_x$ separately. In more detail, the algorithm partitions users into $k$ sets $S_1, S_2, \ldots, S_k$, each with size $n' := \lfloor n/k \rfloor$. For $x \in [k]$, information from users in $S_x$ will be used to estimate $\mathbf{p}_x$. Note that for all $j \in S_x$, $M_j(x)$, the number of times $x$ appears in $X_j$, is a sample from $\text{Bin}(m, \mathbf{p}_x)$. The crux of the algorithm then relies on solving the following problem of estimating the bias of Binomial distributions with one-bit protocols.

**Binomial estimation.** Given $n'$ i.i.d. samples $Z^{n'} = (Z_1, Z_2, \ldots, Z_{n'})$ from $\text{Bin}(m, p)$, the goal is to design one-bit messaging schemes $W_i : [m] \to \{0, 1\}$ and $\hat{p}$, which minimizes

$$\mathbb{E}\left[\left(\hat{p}(Y^{n'}) - p\right)^2\right],$$

where $\forall i \in [n'], Y_i = W_i(Z_i)$.

Without communication constraints, it is folklore that the simple averaging estimator can achieve the optimal estimation error of $O(p(1-p)/mn')$. In this section, we show that there exists a one-bit protocol which achieves this rate with an extra error exponentially small in $n'$, stated below.

**Lemma 2.2.** *Given $n'$ users each observing an i.i.d. sample from $\text{Bin}(m, p)$, where $n' > c \log(m)$ for some absolute constant $c > 0$, there exists a one-bit protocol which outputs an estimate $\hat{p}$ satisfying*

$$\mathbb{E}\left[(\hat{p} - p)^2\right] = O\left(\frac{p(1-p)}{mn'} + \frac{1}{m}e^{-\frac{n'}{240 \log(m)}}\right).$$

We give a high-level overview of the binomial estimation protocol in Section 2.1. The proof of the lemma is rather technical and we defer it to the supplemental. We next explain how the lemma can be used to establish Theorem 2.1. By applying the algorithm of Lemma 2.2 to each $x \in [k]$ with the $n' = \lfloor n/k \rfloor$ users from group $S_x$, we have

$$\mathbb{E}\left[\|\hat{\mathbf{p}} - \mathbf{p}\|_2^2\right] = \sum_{x \in [k]} \mathbb{E}\left[(\hat{\mathbf{p}}_x - \mathbf{p}_x)^2\right] = O\left(\frac{\sum_{x \in [k]} \mathbf{p}_x(1 - \mathbf{p}_x)}{mn'} + \frac{k}{m}e^{-\frac{n'}{240 \log(m)}}\right) = O\left(\frac{k}{mn}\right),$$

where the last bound holds when $n > 100k \log(m) \log(n)$. Applying Cauchy–Schwarz and Jensen's inequalities, we get as desired,

$$\mathbb{E}[\text{TV}(\hat{\mathbf{p}}, \mathbf{p})] \le \frac{1}{2}\sqrt{k\mathbb{E}\left[\|\hat{\mathbf{p}} - \mathbf{p}\|_2^2\right]} = O\left(\sqrt{\frac{k^2}{mn}}\right).$$

### 2.1 Binomial estimation with one-bit protocol

In this section, we provide an overview of the protocol which achieves the guarantee stated in Lemma 2.2. We start by introducing a threshold-based algorithm to provide some motivation.

### 2.1.1 Motivation

We motivate our algorithm by discussing the two naïve algorithms in Section 1.3: why they work in certain regimes and fail in others. We consider the following family of algorithms,

*Fix threshold $t \in [0, m]$: upon observing $Z_i$, user $i$ sends $Y_i = \mathbb{1}\{Z_i \geq t\}$.*

Note that choosing $t = \lceil m/2 \rceil$ and $t = 1$, respectively, gives the two naïve algorithms. Using the above algorithm, one can calculate the empirical estimate of $P_{m,t}(p) := \Pr_{Z \sim \mathrm{Bin}(m,p)}[Z \geq t]$,

$$\hat{P} = \frac{1}{n'} \sum_{i=1}^{n'} Y_i,$$

We then obtain an estimate using $\hat{p} = P_{m,t}^{-1}(\hat{P})$. It can be shown that that $\mathbb{E}\left[|\hat{P} - P_{m,t}(p)|\right] \leq \sqrt{P_{m,t}(p)(1 - P_{m,t}(p))/n'}$ by standard argument of estimating Bernoulli random variables. If $\hat{P}$ is sufficiently close to $P_{m,t}(p)$, we can roughly bound the error for estimating $p$ as

$$\mathbb{E}[|\hat{p} - p|] \approx \frac{1}{|P'_{m,t}(p)|} \mathbb{E}\left[|\hat{P} - P_{m,t}(p)|\right].$$

Hence for better accuracy, we want the derivative $|P'_{m,t}(p)|$ to be large. However, this can only be guaranteed if $p$ and $t/m$ are not too far apart. In fact, to obtain the desired error of $O(p(1-p)/mn')$, we need roughly $|p - t/m| = O(\sqrt{p(1-p)/m})$, the standard deviation of $\mathrm{Bin}(m, p)$.

Hence any fixed threshold-based scheme, including the two naïve algorithms, cannot achieve the optimal rate for all $p$. However, this provides an important intuition for our algorithm: if we can first estimate $p$ up to constant standard deviation, then we can obtain a good estimate with one bit by sending a proper threshold. This motivates the two-stage algorithm in the next section.

### 2.1.2 Algorithm

As outlined in the foregoing informal discussion, the algorithm has two stages, a *localization* stage and a *refinement* stage. Similar structures have also been used in estimating the mean of Gaussian distributions under communication constraints [7, 8]. Note that although the protocol is presented in a two-stage manner, the two stages can be implemented by two disjoint subset of users, who can send the messages simultaneously. Hence the protocol is *noninteractive*. We first introduce Gray codes.

**Definition 2.3** (Gray code). An $s$-bit Gray code $G_s \colon [0, 1] \to \{-1, +1\}^s$ is defined by a collection of Gray functions $g_i \colon [0, 1] \to \{0, 1\}, i \in [s]$, where

$$g_i(x) = \begin{cases} 0 & \lfloor 2^i x \rfloor \mod 4 = 0, 3, \\ 1 & \lfloor 2^i x \rfloor \mod 4 = 1, 2. \end{cases}$$

And the encoding is given by $G_s(x) = (g_1(x), g_2(x), \dots, g_s(x))$.

Gray code is a widely used code in wireless systems to guarantee robustness to bit errors. The code has the following nice property.

**Claim 2.4.** $G_s$ *partitions* $[0, 1]$ *into* $2^s$ *intervals such that, for any two consecutive intervals, the encodings only differ at one bit.*

The property guarantees that if a user can obtain $\hat{p}$ such that $|p - \hat{p}| < 2^{-s}$, then $G_s(\hat{p})$ and $G_s(p)$ only differ in one bit. This is crucial in establishing the performance of the algorithm.

Next we introduce the protocol. We divide users into 4 groups $S_1, \dots, S_4$, each with $N := \frac{n'}{4}$ users. Let $s = \log(m)$ and $M = \frac{N}{s}$.

**Localization.** This stage uses samples from $S_1$ to locate $p$ up to accuracy $\Theta(\sqrt{p(1-p)/m})$ with high probability. Since each user has a sample $Z \sim \mathrm{Bin}(m, p)$, by standard concentration inequalities, $|Z/m - p| \leq \sqrt{2p(1-p)/m}$ with high probability. $S_1$ is further partitioned into $s$ subgroups $(S_1^j)_{j \leq s}$ of equal size. All users $u \in S_1^j$ send the $j$th bit of the Gray code for $Z_u$, *i.e.*, $Y_u = g_j(Z_u/m)$.

The server then obtains majority votes for the $j$th bit $b_j := \mathbb{1}\left\{\sum_{u \in S_1^j} Y_u \geq M/2\right\}$. By decoding $(b_1, \ldots, b_s)$, we obtain a coarse estimate of $p$. By standard concentration analysis and the property of Gray code, it can be shown that with high probability, the estimate has the desired accuracy.

**Refinement.** In this stage, we improve the accuracy of estimating $p$ to $\Theta(\sqrt{p(1-p)/mn'})$ using samples from $S_2, S_3, S_4$. If interactivity is allowed, users in this stage only need to send the threshold function at the crude estimate. To remove interactivity, users in each of the groups will send indicators of whether their sample lies in a certain subset, defined below. Albeit much more involved, these indicator functions are used in a very similar fashion as the threshold functions.

1. We define a partition of $[0, 1]$. Let $C_I$ be a constant and $r := \lfloor\sqrt{\frac{m}{2C_I}}\rfloor$. Define the intervals $I_i := [l_{i-1}, l_i]$ for $1 \leq i \leq r$, where
$$l_i := \min\left\{\frac{C_I i^2}{m}, \frac{1}{2}\right\}, \quad 0 \leq i \leq r.$$
   Furthermore let $I_{2r+1-i} := [1 - l_i, 1 - l_{i-1}]$. User $u \in S_2$ sends $Y_u = \mathbb{1}\{Z_u/m \in \cup_i I_{2i}\}$

2. Let $j_i$ be the midpoint of $I_i$ for $1 \leq i \leq 2r$ and $j_0 = 0, j_{2r+1} = 1$. Define $J_i := [j_{i-1}, j_i]$. User $u \in S_3$ sends $Y_u = \mathbb{1}\{Z_u/m \in \cup_i J_{2i}\}$.

3. User $u \in S_4$ sends $Y_u = \mathbb{1}\{Z_u \geq 1\}$.

To estimate $p$, we show that if $p$ is successfully estimated up to accuracy $\Theta(\sqrt{p(1-p)/m})$ in the localization stage, then for any $p$ within an interval around the estimate, at least one of the expectations of the above indicators has a large derivative with respect to $p$. Hence we show that the server can refine the estimate for $p$ by inverting the corresponding empirical frequency within the interval. Details are provided in the supplementary.

The design of the indicator functions is a keycontribution that sets our algorithm apart from that of [7, 8]. Unlike in the Gaussian case, the variance of a binomial changes with its mean. Note that if $p \in I_i$, $|I_i| = \Theta(\max\{\sqrt{p(1-p)/m}, 1/m\})$. Hence this partition reflects the behavior of the binomial variance as $p$ varies.

## 3 Distribution estimation with $\ell$ bits: a phase transition

In this section, we consider the general setting where each user can send $\ell$ bits. We first note that since each user observes $m$ samples from $[k]$, each multiset of $m$ samples can take at most $N_S = \binom{m+k-1}{m}$ possible values. As it is enough for a protocol to only transmit information about the multiset, or equivalently, the empirical frequencies of the samples, $\log N_S = \Theta(m \log(k/m) + k \log(m/k))$ bits are enough for a user to transmit all their information. Hence here we focus on the case when $\ell = O(m \log(k/m) + k \log(m/k))$.

As mentioned in Section 1.3, we obtain different bounds for different regimes of parameters. When $m < k/2^\ell$, we show in Section 3.1 that the risk scales at rate $1/\sqrt{2^\ell}$ (Theorem 1.1). When $m \geq k/2^\ell$, however, our bounds scale as $1/\sqrt{\ell}$ (Theorem 1.2). In this regime, we consider two different cases: (i) $m < k, \ell > \log(k/m)$; (ii) $m \geq k$, presented in Sections 3.2 and 3.3 separately.

### 3.1 Regime $m \leq k/2^\ell$: an exponential rate in $\ell$

In this subsection, we present an algorithm which achieves the upper bound part of Theorem 1.1. Without loss of generality, we assume each user can send $\ell + 1$ bits as this will only change the resulting bound by a factor of $\sqrt{2}$. Since $2^\ell \leq k/m \leq k$, we cannot represent a symbol in $[k]$ losslessly using $\ell$ bits. Instead, we partition the domain into $t = \lceil k/(2^\ell - 1)\rceil$ blocks $B_1, \ldots, B_t$, each of size at most $2^\ell - 1$. Our protocol builds on the observation that, given a fixed block $B_i$, with $\ell$ bits each user can either represent a symbol in $B_i$ that has appeared in their samples or indicate that none of symbols has appeared.

Before stating the protocol, we first define some related quantities. For $\mathbf{p} \in \Delta_k$, let $\mathbf{p}_B$ be the distribution over the blocks induced by $\mathbf{p}$, where for all $j \in [t]$ $\mathbf{p}_B(j) = \sum_{x \in B_j} \mathbf{p}_x$. For block $j \in [t]$, let $\bar{\mathbf{p}}_j(x)$ be the normalized distribution over elements in set $B_j$, *i.e.*, for all $x \in B_j$, $\bar{\mathbf{p}}_j(x) = \mathbf{p}(x)/\mathbf{p}_B(j)$ (if $\mathbf{p}_B(j) = 0$, we set $\bar{\mathbf{p}}_j(x) = 1/(2^\ell - 1)$, which is the uniform distribution

over $B_j$). The protocol learns $\mathbf{p}$ by estimating both the distribution over blocks and the normalized distributions within each block.

**Estimating $\mathbf{p}_B$.** Each user maps the observed samples to the set they belong to in $B_1, \ldots, B_t$. Then the users use the first bit to run the 1-bit protocol in Theorem 2.1 to learn $\mathbf{p}_B$. The server collects the messages and obtain an estimate $\widehat{\mathbf{p}}_B$.

**Estimating normalized distributions.** Users are partitioned into $t$ sets $S_1, S_2, \ldots, S_t$, each of size $\lfloor n/t \rfloor$.[3] For $i \in S_j$, user $i$ uses the remaining $\ell$ bits to send the first sample it observes in $B_j$. If none of them is observed, the user sends $\perp$.

Upon receiving the messages, the server uses the empirical estimator to estimate $\bar{\mathbf{p}}_j$,

$$\widehat{\mathbf{p}}_j(x) = \frac{\sum_{i \in S_j} \mathbb{1}\{Y_i = x\}}{\sum_{i \in S_j} \mathbb{1}\{Y_i \neq \perp\}}, \qquad x \in [2^\ell],$$

if $\sum_{i \in S_j} \mathbb{1}\{Y_i \neq \perp\} \neq 0$. Otherwise, we set $\widehat{\mathbf{p}}_j(x) = 1/(2^\ell - 1)$ for all $x \in S_j$.

The final estimates are simply

$$\widehat{\mathbf{p}}(x) = \widehat{\mathbf{p}}_B(j) \cdot \widehat{\mathbf{p}}_j(x), \qquad \forall j \in [t], x \in B_j.$$

Next we prove that the estimator achieves the upper bound guarantee in Theorem 1.1. We first relate the estimation errors for $\widehat{\mathbf{p}}$ to those of $\widehat{\mathbf{p}}_B$ and $\widehat{\mathbf{p}}_j$.

**Lemma 3.1.** We have $\mathbb{E}[\mathrm{TV}(\widehat{\mathbf{p}}, \mathbf{p})] \leq \mathbb{E}[\mathrm{TV}(\widehat{\mathbf{p}}_B, \mathbf{p}_B)] + \sum_{j \in [t]} \mathbf{p}_B(j) \mathbb{E}[\mathrm{TV}(\widehat{\mathbf{p}}_j, \bar{\mathbf{p}}_j)]$.

*Proof.* We write

$$\mathbb{E}[\mathrm{TV}(\widehat{\mathbf{p}}, \mathbf{p})] = \frac{1}{2}\mathbb{E}\Big[\sum_{j \in [t]}\sum_{x \in B_j} |\widehat{\mathbf{p}}(x) - \mathbf{p}(x)|\Big] = \frac{1}{2}\mathbb{E}\Big[\sum_{j \in [t]}\sum_{x \in B_j} |\widehat{\mathbf{p}}_B(j)\widehat{\mathbf{p}}_j(x) - \mathbf{p}_B(j)\mathbf{p}_j(x)|\Big]$$

$$\leq \frac{1}{2}\mathbb{E}\Big[\sum_{j \in [t]}\sum_{x \in B_j} (|\widehat{\mathbf{p}}_B(j)\widehat{\mathbf{p}}_j(x) - \mathbf{p}_B(j)\widehat{\mathbf{p}}_j(x)| + |\mathbf{p}_B(j)\widehat{\mathbf{p}}_j(x) - \mathbf{p}_B(j)\mathbf{p}_j(x)|)\Big]$$

$$= \frac{1}{2}\mathbb{E}\Big[\sum_{j \in [t]}\sum_{x \in B_j} \widehat{\mathbf{p}}_j(x)|\widehat{\mathbf{p}}_B(j) - \mathbf{p}_B(j)|\Big] + \frac{1}{2}\mathbb{E}\Big[\sum_{j \in [t]} \mathbf{p}_B(j) \sum_{x \in B_j} |\widehat{\mathbf{p}}_j(x) - \mathbf{p}_j(x)|\Big]$$

$$= \mathbb{E}[\mathrm{TV}(\widehat{\mathbf{p}}_B, \mathbf{p}_B)] + \sum_{j \in [t]} \mathbf{p}_B(j) \mathbb{E}[\mathrm{TV}(\widehat{\mathbf{p}}_j, \bar{\mathbf{p}}_j)]. \qquad \square$$

Next we bound terms in Lemma 3.1 separately. Since $\mathbf{p}_B$ is over a smaller domain of size $\lceil k/(2^\ell - 1) \rceil$, by Theorem 2.1, we have when $n > 100\frac{k}{2^\ell} \log m \log n$,

$$\mathbb{E}[\mathrm{TV}(\widehat{\mathbf{p}}_B, \mathbf{p}_B)] = O\left(\sqrt{\frac{k^2}{mn(2^\ell)^2}}\right). \tag{4}$$

We then bound the error on estimating $\bar{\mathbf{p}}_j$'s. Note that since each user in $S_j$ either sends the first sample it observes in $B_j$ or $\perp$ if none of the elements appears, we have the following facts:

1. For $j \in [t]$, let $N_j := \sum_{i \in S_j} \mathbb{1}\{Y_i \neq \perp\}$, the number of samples the server receives from $B_j$. Then $N_j$ follows a binomial distribution $\mathrm{Bin}(\lfloor n/t \rfloor, \beta_j)$ where $\beta_j := 1 - (1 - \mathbf{p}_B(j))^m = \Theta(\min\{m\mathbf{p}_B(j), 1\})$.

2. For $j \in [t]$, conditioned on that the first sample in $B_j$ is sent, the sample is distributed according to the normalized distribution $\bar{\mathbf{p}}_j$.

Based on these observations, we obtain the following lemma.

**Lemma 3.2.** If $\mathbf{p}_B(j) > 0$, then $\mathbb{E}[\mathrm{TV}(\widehat{\mathbf{p}}_j, \bar{\mathbf{p}}_j)] = O\left(\sqrt{\frac{k}{n\beta_j}}\right)$.

*Proof.* When $\beta_j \leq \frac{8k}{n}$, the bound is trivial since TV distance is always bounded by 1. When $\beta_j > \frac{8k}{n}$, by a Chernoff bound, we have $\Pr\left(N_j < \frac{n\beta_j}{2t}\right) \leq \exp\left(-\frac{n\beta_j}{8t}\right)$. When $N_j \geq \frac{n\beta_j}{2t}$,

$$\mathbb{E}\left[\mathrm{TV}(\widehat{\mathbf{p}}_j, \bar{\mathbf{p}}_j) \mid N_j \geq \frac{n\beta_j}{2t}\right] = O\left(\sqrt{\frac{2^\ell - 1}{N_j}}\right) = O\left(\sqrt{\frac{k}{n\beta_j}}\right).$$

---

[3]If $t$ does not divide $n$, the protocol ignores the last $n - t\lfloor n/t \rfloor$ users.

Combining both, we get

$$\mathbb{E}[\mathrm{TV}(\widehat{\mathbf{p}}_j, \bar{\mathbf{p}}_j)] = O\left(\sqrt{\frac{k}{n\beta_j}} + \exp\left(-\frac{n\beta_j 2^\ell}{8k}\right)\right) = O\left(\sqrt{\frac{k}{n\beta_j}}\right),$$

completing the proof of the lemma. $\qquad\square$

Next we bound the second term in Lemma 3.1 using Lemma 3.2, since $\beta_j = \Theta(\min\{m\mathbf{p}_B(j), 1\})$,

$$\sum_{j\in[t]} \mathbf{p}_B(j)\mathbb{E}[\mathrm{TV}(\widehat{\mathbf{p}}_j, \bar{\mathbf{p}}_j)] = O\left(\sum_{j\in[t]} \max\left\{\mathbf{p}_B(j)\sqrt{\frac{k}{n}}, \sqrt{\frac{\mathbf{p}_B(j)k}{nm}}\right\}\right)$$

$$= O\left(\max\left\{\sqrt{\frac{k}{n}}, \sqrt{\frac{kt}{nm}}\right\}\right). \tag{5}$$

$$= O\left(\sqrt{\frac{k^2}{nm2^\ell}}\right), \tag{6}$$

where (6) follows from $t = \lceil k/(2^\ell - 1)\rceil > m$. Together with (4), this completes the proof.

## 3.2 Regime $m < k$, $\ell > \log(k/m)$: increasing $m$ barely helps

When $m \geq k/2^\ell$, the first term in (5) dominates (up to constant factors) and the bound will not further decrease as $t$ decreases (or $\ell$ increases when using the parameter in Section 3.1). Hence we fix $t = 2m$, where the two terms in (5) match. Similar to the protocol in Section 3.1, we divide the domain into $t = 2m$ non-overlapping blocks $B_1, \ldots, B_t$, each with size $\lceil k/(2m)\rceil$. Then we can define $\mathbf{p}_B$ and $\bar{\mathbf{p}}_j, j \in [t]$ similarly as in Section 3.1 and estimate them separately.

**Estimating $\mathbf{p}_B$.** Users follow the procedures as in Section 3.1 and the server obtains $\widehat{\mathbf{p}}_B$ using the 1-bit protocol from Theorem 2.1.

**Estimating normalized distributions.** Since each block is only of size $\lceil k/(2m)\rceil$, given a block index, each user can send an element within the block with $\log\lceil k/(2m)\rceil < \ell$ bits. To take advantage of this, we assign each user $t' = \lfloor \ell/\log\lceil k/(2m)\rceil\rfloor$ different blocks. More precisely, user $i$ is assigned blocks $B_j$ for $j = (i-1)t' + 1, (i-1)t' + 2, \ldots, it' \mod t$. For each $B_j$, the user sends the first sample they observe in $B_j$ or sends $\perp$ if none of them appears.

In total, $n$ users send out $nt'$ messages, which we index as $(Z_i)_{i\in[nt']}$. Based on which of the $t$ blocks the messages correspond to, we can divide them into $t$ sets where $S_j = \{i \in [n't] \mid j = i \mod t\}$. The server collects the messages from the users and uses the empirical estimator to estimate each normalized distribution within each block, where for $j \in [2t]$, $x \in [[\lceil k/(2m)\rceil]]$,

$$\widehat{\mathbf{p}}_j(x) = \frac{\sum_{i\in S_j} \mathbb{1}\{Z_i = x\}}{\sum_{i\in S_j} \mathbb{1}\{Z_i \neq x\}}.$$

We show the above protocol achieves the upper bound in Theorem 1.2. The proof is in similar spirit as the proof in Section 3.1 with only two differences. First we fix $t = 2m$, hence the first term in (5) dominates. However, since for each block $B_j$, the server receives $|S_j| \geq nt'/(2t) = \Theta(\frac{n\ell}{m\log(k/m+1)})$ users, the estimation error for the normalized distribution within the block can be improved by a factor of $t'$. We defer the details of the proof to the supplemental.

## 3.3 Regime $m \geq k$: learning each $\mathbf{p}_x$ separately

In this case, the protocol in Section 3.2 will not apply since the domain is of size $k < 2m$. We instead consider a protocol which estimates each $\mathbf{p}_x$ using samples from $\mathrm{Bin}(m, \mathbf{p}_x)$ as we did in Section 2. Since now each user has $\ell$ bits, the users can send information about $\ell$ elements, which improves the rate by a $1/\sqrt{\ell}$ factor. We next describe the protocol formally.

We assume without loss of generality $\ell \leq k$; otherwise, each user only uses $k$ bits of its allowed $\ell$.[4] To each $x \in [k]$, we assign a set of users $S_x$ as follows. User $i$ is in set $S_x$ if and only if

---

[4]This is enough since plugging in $\ell = k$ in Theorem 1.2 already achieves the optimal centralized rate.

$\exists i' \in [\ell-1], (i-1)\ell + i' = x \bmod k$. It can be verified that each user is assigned to exactly $\ell$ sets and that $|S_x| \geq \frac{n\ell}{2k}$ for all $x$. Once the sets are assigned, users in $S_x$ run the one-bit protocol in Lemma 2.2 to estimate $\mathbf{p}_x$. By Lemma 2.2, we have when $n\ell > 100k \log m \log n$,

$$\mathbb{E}\left[\|\hat{\mathbf{p}} - \mathbf{p}\|_2^2\right] = \sum_{x \in [k]} \mathbb{E}\left[(\hat{\mathbf{p}}_x - \mathbf{p}_x)^2\right] = O\left(\frac{\sum_{x \in [k]} \mathbf{p}_x(1 - \mathbf{p}_x)}{m \min_x |S_x|} + \frac{k}{m} e^{-\frac{\min_x |S_x|}{30 \log(m)}}\right) = O\left(\frac{k^2}{mn\ell}\right).$$

Hence, applying Cauchy–Schwarz and Jensen's inequalities we get, as desired,

$$\mathbb{E}[\mathrm{TV}(\hat{\mathbf{p}}, \mathbf{p})] \leq \frac{1}{2}\sqrt{k\mathbb{E}\left[\|\hat{\mathbf{p}} - \mathbf{p}\|_2^2\right]} = O\left(\sqrt{\frac{k^2}{mn\ell}}\right).$$

# 4 Lower bounds

In this section, we provide a sketch of the proof for the lower bound parts of Theorem 1.1 and Theorem 1.3. We first use the Poisson sampling trick to show that the problem is at least as hard as estimating the mean of a suitable product of Poisson distributions. Then we leverage the recently developed lower bound technique from [4] to prove minimax estimation lower bounds for that product of Poisson distributions. We provide the complete proof in the supplementary.

**Product of Poisson distributions.** Let $\theta \in \mathbb{R}^k$ be a vector. A product of Poisson distributions $p_\theta$ is a product distribution over $\mathbb{N}^k$ whose $i$th marginal is a $\mathrm{Poi}(\theta(i))$ distribution: $p_\theta = \otimes_{i=1}^k \mathrm{Poi}(\theta(i))$. We next introduce a family of product of Poisson distributions, which will be used as hard instances. For all $z \in \{-1, +1\}^k$, define $\theta_z$ as
$$\theta_z(i) = \frac{m(1 + 2\varepsilon z_i)}{k}, \qquad i \in [k].$$

For $z \in \{-1, +1\}^k$, let $p_z = p_{\theta_z}$ and $\mathcal{P} = \{p_z : z \in \{-1, +1\}^k\}$. Consider now the following generative process: (1) Pick $Z$ uniformly from $\{-1, +1\}^k$. (2) Users observe samples $X^n$ from $\mathbf{p}_Z^{\otimes n}$. (3) Users follow a protocol $\Pi$ and send messages $Y^n$. The following lemma states that if there exists a discrete distribution learning protocol, there must exist a protocol which is able to extract enough information about $Z$ from $Y^n$.

**Lemma 4.1** (Assouad-type bound). *If there exists a discrete distribution learning protocol with $m$ samples per user which has expected TV error at most $\varepsilon/10$, there must exist a protocol $\Pi$ whose messages $Y^n$ under the above defined process satisfy $\sum_{i \in [k]} I(Z_i; Y^n) = \Omega(k)$.*

The proof combines standard lower bound techniques of Poisson sampling and Assouad's Lemma, which we present in the supplementary. The challenging part of the argument then lies in bounding the sum of mutual informations in Lemma 4.1 for communication-constrained protocols, which we do in the next lemma:

**Lemma 4.2.** *Let $Z - X^n - Y^n$ be the Markov chain defined above. For any $\ell$-bit interactive protocol:*

1. *When $m2^\ell < k$, we have $\sum_{i \in [k]} I(Z_i; Y^n) \leq \frac{mn2^\ell \varepsilon^2}{k}$.*

2. *When $m > k \log k$ and $n < (k/\ell)^2$, we have $\sum_{i \in [k]} I(Z_i; Y^n) \leq \frac{mn\ell \varepsilon^2}{k}$.*

We provide the detailed proof of these two points in the supplementary, and only provide a sketch here. The bounds are proved using the $\chi^2$-contraction bounds recently developed in [4]. The bounds relates the amount of information to a function $\phi_{z,i} \propto (\frac{d\mathbf{p}_{z \oplus i}}{d\mathbf{p}_z} - 1)$, where $z^{\oplus i}$ denotes the vector obtained by flipping the $i$th coordinate of $z$. Obtaining the bound $\sum_{i \in [k]} I(Z_i; Y^n) = O(2^\ell)$, while quite involved, then can be done using ideas from [4]. Getting the tight bound of $\sum_{i \in [k]} I(Z_i; Y^n) = O(\ell)$ when $m > k \log k$, however, requires some additional ideas. Indeed, to get such a dependence on $\ell$, one would typically argue about and use the subgaussian tail of $\phi_{z,i}$; unfortunately, the quantity $\phi_{z,i}$ provably does *not* exhibit such tails for the family of Poisson distributions. To circumvent this issue, we decompose $\phi_{z,i}$ into a subgaussian part and a remainder part, and bound each term separately (noting that the subgaussian part only dominates in the regime $m \gg k$).

Finally, we show how the lemmas can be used to derive the lower bound of Theorem 1.1. Assume there exists an $\ell$-bit protocol which estimates any discrete distribution up to expected TV error $\varepsilon$. By Lemma 4.2 (Item 1) and Lemma 4.1, we have $\frac{mn2^\ell \varepsilon^2}{k} \geq \sum_{i \in [k]} I(Z_i; Y^n) = \Omega(k)$.

Reorganizing, this implies $\varepsilon = \Omega(\sqrt{\frac{k^2}{mn2^\ell}})$, as stated in Theorem 1.1. Theorem 1.3 can be obtained similarly using Lemma 4.2 (Item 2) and Lemma 4.1.

## Acknowledgments and Disclosure of Funding

Jayadev Acharya is supported by NSF-CCF- 1846300 (CAREER), NSF-CCF-1815893, and a Google Faculty Fellowship. Yuhan Liu is supported by AFRI Competitive Grant no. 2020-67021-32855/project accession no. 1024262 from the USDA National Institute of Food and Agriculture, and NSF-CCF- 1846300 (CAREER). Ziteng Sun is supported by NSF-CCF- 1846300 (CAREER). Himanshu Tyagi is supported in part by a research grant from the Robert Bosch Center for Cyberphysical Systems (RBCCPS), Indian Institute of Science, Bangalore.

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
