# Appendix

## Table of Contents

## A   Binomial estimation with 1 bit: proof of Lemma 2.2

We first provide the detailed algorithm for binomial estimation. Before doing so we recall a few definitions. We start with the notion of distance between a point and a set: for $A \subseteq [0,1]$ and $x \in [0,1]$, let $d(x, A) := \inf_{y \in A} |x - y|$.

Our algorithms will crucially rely on the use of *Gray codes*, which we define next.

**Definition A.1** (Gray code)**.** An $s$-bit Gray code $G_s \colon [0,1] \to \{-1, +1\}^s$ is defined by a collection of Gray functions $g_i \colon [0,1] \to \{-1, +1\}, i \in [s]$, where

$$g_i(x) = \begin{cases} -1 & \text{if } \lfloor 2^i x \rfloor \bmod 4 \in \{0, 3\}, \\ 1 & \text{if } \lfloor 2^i x \rfloor \bmod 4 \in \{1, 2\}. \end{cases}$$

The encoding is then given by $G_s(x) := (g_1(x), g_2(x), \ldots, g_s(x))$.

Accordingly, we define the decoding function for an $s$-bit gray code:

$$\mathrm{Dec}_s(b_1, \ldots, b_s) := \{x \in [0,1] \colon g_j(x) = b_j, j = 1, \ldots, s\}. \tag{7}$$

It can be shown [8] that, for all $(b_1, \ldots, b_s) \in \{-1, +1\}^s$, there exist $0 \leq a < b \leq 1$ satisfying $b - a = 2^{-s}$ and $\mathrm{Dec}_s(b_1, \ldots, b_s) = (a, b]$.

Let $C_I$ be a constant and $r := \lfloor \sqrt{\frac{m}{2C_I}} \rfloor$. We now recall the definition of the intervals $\{I_i\}_{i \in [2r]}$ and $\{J_i\}_{i \in [2r+1]}$. Let $I_i := [l_{i-1}, l_i]$ for $1 \leq i \leq r$, where

$$l_i := \min\left\{\frac{C_I i^2}{m}, \frac{1}{2}\right\}, \quad 0 \leq i \leq r.$$

Furthermore $I_{2r+1-i} := [1 - l_i, 1 - l_{i-1}]$. Let $j_i$ be the midpoint of $I_i$ for $1 \leq i \leq 2r$ and $j_0 = 0, j_{2r+1} = 1$. Define $J_i := [j_{i-1}, j_i], i \in [2r + 1]$.

It can be seen that both $\{I_i\}_{i \in [2r]}$ and $\{J_i\}_{i \in [2r+1]}$ are partitions of $[0, 1]$. The length of each interval is chosen such that $\forall i \in [2r]$ and $p \in I_i$, we have $|I_i| \asymp \max\{\sqrt{p/m}, 1/m\}$ (up to multiplicative constants), which is at least as large as the standard deviation of $Z/m$ if $Z \sim \mathrm{Bin}(m, p)$. Our algorithm then builds on the fact that with high probability, $Z/m$ falls into either the interval $p$ belongs to or one of its two adjacent intervals. The same statement holds for the $J_i$'s.

For $p \in [0, 1]$, define the functions

$$R_2(p) := \Pr\left(\frac{Z}{m} \in \bigcup_i I_{2i}\right), \quad R_3(p) := \Pr\left(\frac{Z}{m} \in \bigcup_i J_{2i}\right), \quad R_4(p) = \Pr\left(Z \geq 1\right), \tag{8}$$

where $Z \sim \mathrm{Bin}(m, p)$. For an interval $I$, we define $R_{i,I}, i = 2, 3, 4$ as the restriction of $R_i$ on the interval $I$. We next describe the detailed protocol below.

---
**One-bit Binomial Estimation Protocol.**

Divide users into 4 groups $S_1, \ldots, S_4$, each with size $N := \frac{n'}{4}$. Let $s = \log(m)$, $M = \frac{N}{s}$.

**Localization stage.** In this stage, the goal is to obtain an interval $I$, which corresponds to a crude estimate of $p$.

- **Users**: $S_1$ is further partitioned into $s$ subgroups $(S_1^j)_{j \leq s}$ of equal size. All users $u \in S_1^j$ send the $j$th bit of the Gray code for $Z_u$, *i.e.*, $Y_u = g_j(Z_u/m)$.
- **The server**: For each $j \in [s]$, let $b_j := \mathbb{1}\left\{ \sum_{u \in S_1^j} Y_u \geq M/2 \right\}$ and decode

$$I := \mathrm{Dec}_s(b_1, \ldots, b_s).$$

**Refinement stage.** In this stage, we improve the accuracy to $\Theta(\sqrt{p(1-p)/mn'})$.

- **Users**:
  1. User $u \in S_2$ sends $Y_u = \mathbb{1}\{Z_u/m \in \cup_i I_{2i}\}$.
  2. User $u \in S_3$ sends $Y_u = \mathbb{1}\{Z_u/m \in \cup_i J_{2i}\}$.
  3. User $u \in S_4$ sends $Y_u = \mathbb{1}\{Z_u \geq 1\}$.
- **The server**: One of the 3 following cases must hold.

  **If** $I \subseteq [0, 65 C_I/m]$, let $\bar{Y}_4 = \frac{1}{N} \sum_{u \in S_4}^N Y_u$

  $$\hat{p} = R_4^{-1}\left(\bar{Y}_4\right) := \left\{ p \in [0,1] : R_4(p) = \bar{Y}_4 \right\}.$$

  **Else if** there exists $i \in [2r]$ such that $I \subseteq I_i' := \left[ l_i - \frac{0.55 C_I i}{m}, l_i + \frac{0.55 C_I i}{m} \right]$, let $\bar{Y}_2 = \frac{1}{N} \sum_{u \in S_2}^N Y_u$

  $$\hat{p} = R_{2,I_i'}^{-1}\left(\bar{Y}_2\right) := \left\{ p \in I_i' : R_2(p) = \bar{Y}_2 \right\}.$$

  **Else if** there exists $i \in [2r+1]$ such that $I \subseteq J_i' := \left[ j_i - \frac{0.55 C_I i}{m}, j_i + \frac{0.55 C_I i}{m} \right]$, let $\bar{Y}_3 = \frac{1}{N} \sum_{u \in S_3}^N Y_u$

  $$\hat{p} = R_{3,J_i'}^{-1} := \left\{ p \in J_i' : R_3(p) = \bar{Y}_3 \right\}.$$
---

**Overview.** Next we prove that the above protocol achieves the performance described in Lemma 2.2. The proof requires two steps. First we show that for the localization stage, the expected error due to failing to locate $p$ is small, formally stated as Theorem A.2.

**Theorem A.2.** *Let* $J = \left\{ x : d(x, \mathrm{Dec}_K(b_1, \ldots, b_K)) \leq 8 \max\{\frac{p(1-p)}{m}, \frac{1}{m}\} \right\}$. *Then there exists a constant* $C_{\mathrm{loc}}$ *such that*

$$\mathbb{E}\left[(\hat{p} - p)^2 \mathbb{1}\{p \notin J\}\right] \leq C_{\mathrm{loc}} \max\left\{ \frac{1}{m^2}, \frac{p}{m} \right\} e^{-\frac{n'}{240 \log(m)}}.$$

Then for the refinement stage, we show that conditioned on $p$ is localized successfully, the expected error is small, formally stated as the theorem below.

**Theorem A.3.** *Let* $J$ *be the interval defined in Theorem A.2. Then*

$$\mathbb{E}\left[(\hat{p} - p)^2 \mathbb{1}\{p \in J\}\right] \leq O\left(\frac{p}{mn'}\right).$$

The proof idea is that $J$ is contained in an interval where at least one of $R_2(p)$, $R_3(p)$, $R_4(p)$ is monotonic and has derivative with high magnitude. Hence inverting the corresponding function yields small estimation error.

Combining the two parts naturally implies the desired error rate in Lemma 2.2 (with a slightly different constant in the exponential part).

Next we present the proof of Theorem A.2 in Appendix A.1 and the proof of Theorem A.3 in Appendix A.2. Throughout this section we assume that $m \geq 300$ and $n' \geq 960 \log(m) \log(n'k)$.[5]

## A.1 Localization: proof of Theorem A.2

The proof relies on the following lemmas which guarantee that $p$ is close to the decoded interval at the localization stage with high probability.

**Lemma A.4.** *Let $p \in [0, 1]$ and $1 \leq K \leq \log(\min\{m, \sqrt{m/p(1-p)}\})$. Then there exists a constant $C_1 > 0$ such that for any $L \leq K$,*

$$\Pr[\, d(p,\ \mathrm{Dec}_K(b_1, \dots, b_K)) \geq \frac{5}{4} 2^{-L} - 2^{-K} \,]$$

$$\leq C_1 \left( \exp\left( -\frac{m 2^{-2(L+2)}}{4 \min\{p, 1-p\}} \right) + \exp\left( -\frac{m \max\{\min\{p, 1-p\}, 2^{-(L+2)}\}}{4} \right) \right).$$

**Lemma A.5.** *Let $p \in [0, 1]$ and $1 \leq K \leq \log(\min\{m, \sqrt{m/p(1-p)}\}) - 5$. Then for all $L \leq K$,*

$$\Pr\left[ d(p, \mathrm{Dec}_K(b_1, \dots, b_K)) \geq \frac{5}{4} 2^{-L} - 2^{-K} \right] \leq L \exp\left(-M/240\right).$$

We now show how to prove Theorem A.2 using the lemmas above, and defer the proofs of these lemmas to later sections. Denote $A = d(p, \mathrm{Dec}_K(b_1, \dots, b_K))$.

Furthermore define $N'$ such that $M = 120 \log N'$. Due to the assumption that $n' \geq 480 \log(m) \log(n'k)$, we have $M = N/\log m \geq 120 \log(n'k)$, and hence $N' \geq n'k \geq \log m$. Let $K = \log(\min\{m, \sqrt{m/p(1-p)}\})$ and $K' = \max\{K - \log N', 0\}$.

$$\mathbb{E}\left[A^2 \mathbb{1}\{p \notin J\}\right] \leq \mathbb{E}\left[A^2 \mathbb{1}\left\{A \geq \frac{5}{4} 2^{-K'} - 2^{-K}\right\}\right] + \mathbb{E}\left[A^2 \mathbb{1}\left\{\frac{1}{4} 2^{-K} \leq A \leq \frac{5}{4} 2^{-K'} - 2^{-K}\right\}\right].$$

We bound the first term. If $K' = 0$, then since we always have $A \leq 1 - 2^{-K}$, clearly

$$\mathbb{E}\left[A^2 \mathbb{1}\left\{A \geq \frac{5}{4} 2^{-K'} - 2^{-K}\right\}\right] = 0.$$

Else, $K' = K - \log N'$. Applying Lemma A.4,

$$\mathbb{E}\left[A^2 \, \mathbb{1}\left\{A \geq \frac{5}{4} 2^{-K'} - 2^{-K}\right\}\right]$$

$$\leq \sum_{j=0}^{K'-1} \Pr\left[\frac{5}{4} 2^{-K'+j} - 2^{-K} \leq A \leq \frac{5}{4} 2^{-K'+j+1} - 2^{-K}\right] \left(\frac{5}{4} 2^{-K'+j+1}\right)^2$$

$$\leq \sum_{j=0}^{K'-1} \Pr\left[A \geq \frac{5}{4} 2^{-K'+j} - 2^{-K}\right] \left(\frac{5}{4} 2^{-K'+j+1}\right)^2$$

$$\leq \frac{25}{16} 2^{-2K'} \sum_{j=0}^{K'-1} 2^{2j+2} C_1 \left(\exp\left(-\frac{m 2^{-2(K'-j+2)}}{4 \min\{p, 1-p\}}\right) + \exp\left(-\frac{m 2^{-(K'-j+2)}}{4}\right)\right).$$

Since $K' \leq \log(\min\{m, \sqrt{m/p(1-p)}\})$, we have that for $N' \geq 16$,

$$\exp\left(-\frac{m 2^{-2(K'-j+2)}}{4 \min\{p, 1-p\}}\right) \leq \exp\left(-\frac{m 2^{2(j+K-K'-2-\log(\sqrt{m/p(1-p)}))}}{4 \min\{p, 1-p\}}\right)$$

$$\leq \exp\left(-2^{2j-7} \cdot N'^2\right)$$

$$\leq \exp\left(-2^{2j}\right) \exp\left(-N'^2/128\right).$$

---

[5]The assumption is satisfied for both Theorem 1.1 and Theorem 1.2

and

$$\exp\left(-\frac{m2^{-(K'-j+2)}}{4}\right) \le \exp\left(-2^{j-4}N'\right) \le \exp\left(-2^j\right)\exp(-N'/16).$$

Note that $S := \sum_{j=0}^{\infty} 2^{2j+2}\left(\exp\left(-2^{2j}\right) + \exp\left(-2^j\right)\right) < \infty$. Hence, letting $C_2 := \frac{25}{16}(1 + C_1 S)$, we get that if $N' \ge 100$,

$$\mathbb{E}\left[A^2 \mathbb{1}\left\{A \ge \frac{5}{4}2^{-K'} - 2^{-K}\right\}\right] \le \max\left\{\frac{1}{m^2}, \frac{p(1-p)}{m}\right\} N'^2\left(\exp\left(-\frac{N'}{16}\right) + \exp\left(-\frac{N'^2}{128}\right)\right)$$

$$\le 1024 \cdot \frac{25}{16}\max\left\{\frac{1}{m^2}, \frac{p}{m}\right\}\frac{1}{N'}.$$

For the second term, we have

$$\mathbb{E}\left[A^2 \mathbb{1}\left\{\frac{1}{4}2^{-K} \le A \le \frac{5}{4}2^{-K'-2^{-K}}\right\}\right] \le 1024 \cdot \frac{25}{16}\max\left\{\frac{1}{m^2}, \frac{p(1-p)}{m}\right\} N'^2 \frac{K}{N'^4}$$

$$\le 1024 \cdot \frac{25}{16}\max\left\{\frac{1}{m^2}, \frac{p(1-p)}{m}\right\}\frac{K}{N'^2}$$

$$\le 1024 \cdot \frac{25}{16}\max\left\{\frac{1}{m^2}, \frac{p(1-p)}{m}\right\}\frac{1}{N'},$$

where final inequality is due to the assumption that $M = N/\log m \ge 120\log(n'k)$, and hence $N' \ge n'k \ge \log m$. Combining the two parts yields

$$\mathbb{E}\left[A^2 \mathbb{1}\{p \notin J\}\right] \le C_3'\max\left\{\frac{1}{m^2}, \frac{p}{m}\right\}\frac{1}{N'}.$$

where $C_3' = 3200$. Finally,

$$\mathbb{E}\left[(\hat{p} - p)^2 \mathbb{1}\{p \notin J\}\right] \le \mathbb{E}\left[(A + 2^{-K})^2 \mathbb{1}\{p \notin J\}\right]$$

$$= \mathbb{E}\left[A^2 \mathbb{1}\{p \notin J\}\right] + 2 \cdot 2^{-K}\mathbb{E}[A\mathbb{1}\{p \notin J\}] + 2^{-2K}\Pr[p \notin J]$$

$$\le \mathbb{E}\left[A^2 \mathbb{1}\{p \notin J\}\right] + 2 \cdot 2^{-K}\sqrt{\mathbb{E}[A^2]} + 2^{-2K}\Pr[p \notin J]$$

$$\le C_3'\max\left\{\frac{1}{m^2}, \frac{p}{m}\right\}\frac{1}{N'} + 64\max\left\{\frac{1}{m^2}, \frac{p}{m}\right\}\sqrt{\frac{C_3'}{N'}} + 1024\max\left\{\frac{1}{m^2}, \frac{p}{m}\right\}Ke^{-\frac{M}{30}}$$

$$\le C_{\text{loc}}\max\left\{\frac{1}{m^2}, \frac{p}{m}\right\}e^{-\frac{M}{240}},$$

where $C_{\text{loc}} := C_3' + 64\sqrt{C_3'} + 1024$. $\qquad\square$

### A.1.1 Proof of Lemma A.4

Let $G_j$ be the set of change points of $g_j$,

$$G_j := \{(2i - 1)2^{-j} : 1 \le i \le 2^{j-1}\},$$

that is, the collection of points where $g_j$ changes from 0 to 1 or from 1 to 0. We first bound the probability of error for each Gray code sent by a single user.

**Lemma A.6.** *For user $u \in S_1^j$, recall that $Y_u = g_j(Z_u/m)$ where $Z_u \sim \text{Bin}(m, p)$. We then have*

$$\Pr[Y_u \ne g_j(p)] \le 2\exp\left(-\frac{md(p, G_j)^2}{2(\min\{p, 1-p\} + d(p, G_j))}\right).$$

*Proof.* The proof follows by a Chernoff Bound. For $p \leq 1/2$,

$$\Pr[Y_u \neq g_j(p)] \leq \Pr\left[\frac{Z_u}{m} \leq p - d(p, G_j)\right] + \Pr\left[\frac{X}{m} \geq p + d(p, G_j)\right]$$

$$\leq \exp\left(-mD(p - d(p, G_j)\|p)\right) + \exp\left(-mD_{KL}(p + d(p, G_j)\|p)\right)$$

$$\leq \exp\left(-\frac{md(p, G_j)^2}{2p}\right) + \exp\left(-\frac{md(p, G_j)^2}{2(p + d(p, G_j))}\right)$$

$$\leq 2\exp\left(-\frac{md(p, G_j)^2}{2(p + d(p, G_j))}\right).$$

The case $p \geq 1/2$ follows by symmetry. $\qquad\square$

Since majority vote decreases the error for $M \geq 3$, the upper bound in Lemma A.6 also holds for $\Pr[b_j \neq g_j(p)]$.

We assume $p \leq 1/2$ since $p \geq 1/2$ then directly follows by symmetry. We first make the following observations for $L \geq 2$:

1. There exists a unique $1 \leq L' \leq L$ such that
$$d(p, G_L) + d(p, G_{L'}) = 2^{-L}.$$
This is because $\cup_{j=1}^{L} G_j = \{x \in [0, 1] : x = k/2^L, k = 1, \dots, 2^L - 1\}$. Furthermore, $d(p, G_L) \leq 2^{-L}$, so there must exist a unique $L'$ such that $d(p, G_L) + d(p, G_{L_0}) = 2^{-L}$.

2. For $l \in \mathbb{N}$, there exists at most one $k_l$ such that
$$d(p, G_{k_l}) = d(p, G_L) + l2^{-L}.$$
Similar argument holds with $d(p, G_L)$ replaced by $d(p, G_{L'})$.

If $d(p, \text{Dec}_K(b_1, \dots, b_K)) \geq \frac{5}{4}2^{-L} - 2^{-K})$, then there must exist at least one $j \in [L]$ such that $b_j \neq g_j(p)$. Otherwise there would exist an interval $I$ of length $2^{-L}$ such that $p \in I$ and $\text{Dec}_K(b_1, \dots, b_K) \subseteq I$, which implies $d(p, \text{Dec}_K(b_1, \dots, b_K)) \leq 2^{-L} - 2^{-K}$, contradiction. We bound the probability by a union bound. We consider two cases.

1. $d(p, G_L) \in [\frac{1}{4}2^{-L}, \frac{3}{4}2^{-L}]$. In this case, we also have $d(p, G_{L'}) \in [\frac{1}{4}2^{-L}, \frac{3}{4}2^{-L}])$. Hence,

$$\Pr\left[d(p, \text{Dec}_K(b_1, \dots, b_K)) \geq \frac{5}{4}2^{-L} - 2^{-K}\right]$$

$$\leq \sum_{j=1}^{L} \Pr[b_j \neq g_j(p)]$$

$$\leq \sum_{j \leq L : d(x, G_j) \leq p} 2\exp\left(-\frac{1}{4p}md(p, G_j)^2\right) + \sum_{j \leq L : d(x, G_j) > p} 2\exp\left(-\frac{1}{4}md(p, G_j)\right)$$

$$\leq \sum_{l=0}^{\infty} 2\exp\left(-\frac{1}{4p}m(d(p, G_L) + l2^{-L})^2\right) + \sum_{l=0}^{\infty} 2\exp\left(-\frac{1}{4p}m(d(p, G_{L'})^2 + l2^{-L})\right)$$

$$+ 2\sum_{l=0}^{\infty} 2\exp\left(-\frac{1}{4}m\left(\max\{p, \frac{1}{4}2^{-L}\} + l2^{-L}\right)\right).$$

We bound the terms separately.

$$\sum_{l=0}^{\infty} 2\exp\left(-\frac{1}{4p}m(d(p, G_L) + l2^{-L})^2\right) \leq \sum_{l=0}^{\infty} 2\exp\left(-\frac{1}{4p}m(d(p, G_L)^2 + 2ld(p, G_L)2^{-L})\right)$$

$$\leq 2\exp\left(-\frac{1}{4p}md(p, G_L)^2\right)\frac{1}{1 - \exp(-\frac{1}{4p}m2^{-(2L+1)})}$$

$$\leq \frac{2}{1 - \exp(-1/8)}\exp\left(-\frac{1}{4p}m2^{-2(L+2)}\right).$$

The bound is the same for the second term. For the last term,

$$2\sum_{l=0}^{\infty} 2\exp\left(-\frac{1}{4}m\left(\max\{p,\frac{1}{4}2^{-L}\}+l2^{-L}\right)\right) \leq \frac{4\exp\left(-\frac{1}{4}m\max\{p,2^{-(L+2)}\}\right)}{1-\exp\left(-\frac{1}{4}m2^{-L}\right)}$$

$$\leq \frac{4\exp\left(-\frac{1}{4}m\max\{p,2^{-(L+2)}\}\right)}{1-\exp\left(-1/4\right)}.$$

Combining all the terms yields the desired bound.

2. $d(p,G_L) \in [0,2^{-(L+2)}) \cup (3\cdot 2^{-(L+2)}, 2^{-L}]$

If $d(p,G_L) \in [0,2^{-(L+2)})$, then to ensure $d(p,\mathrm{Dec}_K(b_1,\ldots,b_K)) \geq \frac{5}{4}2^{-L}-2^{-K}$, at least one $j \in [L-1]$ must satisfy $b_j \neq g_j(p)$. If otherwise, let $x_L = \arg\min\{x \in G_L : d(p,x)\}$, then we would have $p \in [x_L - 2^{-L}, x_L + 2^{-L}]$ and $\mathrm{Dec}_K(b_1,\ldots,b_K) \subseteq [x_L - 2^{-L}, x_L + 2^{-L}]$. Hence

$$d(p,\mathrm{Dec}_K(b_1,\ldots,b_K)) < d(p,G_L) + 2^{-L} - 2^{-K} \leq \frac{5}{4}2^{-L} - 2^{-K},$$

which leads to a contradiction.

If $d(p,G_L) \in (3\cdot 2^{-(L+2)}, 2^{-L}]$, then $d(p,G_{L'}) < 2^{-(L+2)}$. Similar to the previous case, at least one $j \in [L]\setminus[L']$ must satisfy $b_j \neq g_j(p)$. Otherwise let $x_{L'} = \arg\min\{x \in G_{L'} : d(p,x)\}$, the remaining argument follows with $x_L$ replaced by $x_{L'}$.

For both cases, we proceed using a union bound. We only present the proof for $d(p,G_L) \in [0,2^{-(L+2)})$.

$$\Pr\left[\, d(p,\mathrm{Dec}_K(b_1,\ldots,b_K)) \geq \frac{5}{4}2^{-L} - 2^{-K}\right]$$

$$\leq \sum_{j=1}^{L-1} \Pr[\, b_j \neq g_j(p)\,]$$

$$\leq \sum_{j\leq L-1 : d(x,G_j)\leq p} 2\exp\left(-\frac{1}{4p}md(p,G_j)^2\right) + \sum_{j\leq L-1 : d(x,G_j)>p} 2\exp\left(-\frac{1}{4}md(p,G_j)\right)$$

$$\leq \sum_{l=1}^{\infty} 2\exp\left(-\frac{1}{4p}m(d(p,G_L)+l2^{-L})^2\right) + \sum_{l=0}^{\infty} 2\exp\left(-\frac{1}{4p}m(d(p,G_{L'})^2+l2^{-L})\right)$$

$$+ 2\sum_{l=0}^{\infty} 2\exp\left(-\frac{1}{4}m\left(\max\{p,\frac{3}{4}2^{-L}\}+l2^{-L}\right)\right)$$

Each term is bounded analogously to the $d(p,G_L) \in [\frac{1}{4}2^{-L}, \frac{3}{4}2^{-L}]$ case, and obtain the same final upper bound.

Combining the two cases completes the lemma. $\qquad\square$

### A.1.2 Proof of Lemma A.5

For each $j \leq \log m$, recall that we let $M$ users vote on $g_j(p)$.

**Lemma A.7.** *Let $b_1,\ldots,b_s$ be the majority votes from the localization stage. Fixed $j \leq s$ and $u \in S_1^j$, let $p_j = \Pr[Y_u \neq\neq g_j(p)]$ and $\gamma_j = 1/2 - p_j$. Recalll that $|S_1^j| = M$. Then,*

$$\Pr[\, b_j \neq g_j(p)\,] \leq \exp\left(-\frac{M}{1+2\gamma_j}\gamma_j^2\right)$$

*Proof.* The lemma follows directly from a Chernoff Bound, since

$$\Pr[\, b_j \neq g_j(p)\,] = \Pr\left[\frac{1}{M}\sum_{u\in S_1^j} \mathbb{1}\{Y_u \neq g_j(p)\} \geq \frac{1}{2}\right] \leq \exp\left(-\frac{M}{2(1-p_j)}\gamma_j^2\right) \qquad\square$$

We once more assume without loss of generality that $p \leq 1/2$, as the case $p \geq 1/2$ follows by symmetry. Similarly to Lemma A.4, we have two cases:

1. $d(p, G_L) \in [\frac{1}{4}2^{-L}, \frac{3}{4}2^{-L}]$. In this case, we also have $d(p, G_{L'}) \in [\frac{1}{4}2^{-L}, \frac{3}{4}2^{-L}])$. Fix $j \leq L$ and $u \in S_1^j$.
   If $p \leq d(p, G_j)$,

   $$\Pr[Y_u \neq g_j(p)] \leq 2\exp\left(-\frac{md(p, G_j)^2}{2(p + d(p, G_j))}\right) \leq 2\exp\left(-\frac{md(p, G_j)}{4}\right) \leq 2\exp\left(-\frac{m2^{-(L+2)}}{4}\right),$$

   and that last quantity is at most $2e^{-2}$.
   If $p > d(p, G_j)$,

   $$\Pr[Y_u \neq g_j(p)] \leq 2\exp\left(-\frac{md(p, G_j)^2}{2(p + d(p, G_j))}\right) \leq 2\exp\left(-\frac{md(p, G_j)^2}{4p}\right) \leq 2\exp\left(-\frac{m2^{-2(L+2)}}{4p}\right),$$

   which is again at most $2e^{-2}$.
   Hence $\gamma_k^2/(1 + 2\gamma_k) \geq 0.072$. Using the proof from Lemma A.4

   $$\Pr\left[d(p, \mathrm{Dec}_K(b_1, \ldots, b_K)) \geq \frac{5}{4}2^{-L} - 2^{-K}\right] \leq \sum_{j=1}^{L} \Pr[b_j \neq g_j(p)]$$
   $$\leq \sum_{j=1}^{L} 2\exp(-M/30)$$
   $$= L\exp(-M/30).$$

2. $d(p, G_L) \in [0, 2^{-(L+2)}) \cup (3 \cdot 2^{-(L+2)}, 2^{-L}]$
   Analogously to Lemma A.4, we either have at least one $j \in \mathcal{S}$ where $\mathcal{S} = [L] \setminus \{L\}$ or $\mathcal{S} = [L] \setminus \{L'\}$ such that $b_j \neq g_j(p)$. Furthermore, $\gamma_j \geq 2/3$ for $j \in \mathcal{S}$. Hence,

   $$\Pr\left[d(p, \mathrm{Dec}_K(b_1, \ldots, b_K)) \geq \frac{5}{4}2^{-L} - 2^{-K}\right] \leq \sum_{j \in \mathcal{S}} \Pr[b_j \neq g_j(p)].$$

   Proceeding similarly as in the first case yields the desired result. $\qquad \square$

## A.2   Refinement: proof of Theorem A.3

We only prove it for $p \leq 1/2$ as the case when $\geq 1/2$ follows by symmetry. Recall $J = \{x \mid d(x, \mathrm{Dec}_K(b_1, \ldots, b_K)) \leq 8\max\{\frac{p(1-p)}{m}, \frac{1}{m}\}\}$, throughout this section we condition on the event that $p \in J$. To prove the performance, we consider following three cases,

**Lemma A.8.** *At least one of the following must hold,*

1. *There exists $i \in [2r]$, such that $J \subseteq I_i' = \left[\frac{C_I i^2}{m} - \frac{0.55 C_I i}{m}, \frac{C_I i^2}{m} + \frac{0.55 C_I i}{m}\right]$*
2. *There exists $i \in [2r+1]$ such that $J \subseteq J_i' = \left[j_i - \frac{0.55 C_I i}{m}, j_i + \frac{0.55 C_I i}{m}\right]$*
3. *$J \subseteq [0, 65C_I/m]$*

We provide the proof in Appendix A.2.1. Recall from the protocol that for Case 1, we invert $R_2(p)$ in the corresponding interval. For Case 2, we invert $R_3(p)$. For Case 3, we invert $R_4(p)$.

For $i \in \{2, 3, 4\}$, recall that $\bar{Y}_i = \frac{1}{N}\sum_{u \in S_i} Y_u$. Since $\forall u \in S_i$ $Y_u$ is a Bernoulli random variable with bias $R_i(p)$, by a standard variance analysis, we can show that

$$\mathbb{E}\left[(\bar{Y}_i - R_i(p))^2\right] \leq \frac{R_i(p)}{N}.$$

Our main goal is to prove Lemma A.9 which guarantees that these functions have sufficiently large derivatives, and hence that we can define inverse functions and guarantee small estimation error.

**Lemma A.9.** *There exists some absolute constant $C > 0$ such that the following holds.*

1. *For all $i \in [2r]$, $R_2(p)$ is monotonic in $I_i' := \left[l_i - \frac{0.55 C_I i}{m}, l_i + \frac{0.55 C_I i}{m}\right]$, and for $p \in I_i'$,*

$$\left|\frac{dR_2(p)}{dp}\right| \geq C\sqrt{\frac{m}{p}}.$$

2. *For all $i \in [2r+1]$, $R_3(p)$ is monotonic in $J_i' := \left[j_i - \frac{2 C_I i}{m}, j_i + \frac{0.55 C_I i}{m}\right]$, and for $p \in J_i'$,*

$$\left|\frac{dR_3(p)}{dp}\right| \geq C\sqrt{\frac{m}{p}}.$$

3. *$R_4(p)$ is monotonic in $[0, 65 C_I/m]$, and for $p \in [0, 65 C_I/m]$,*

$$\frac{dR_4(p)}{dp} \geq Cm.$$

The proof of this lemma is provided in Appendix A.2.2. We now proceed to prove Theorem A.3 based on Lemmas A.8 and A.9.

If Case 1 holds in Lemma A.8, we have

$$\mathbb{E}\left[(\hat{p}-p)^2 \,|p \in J\right] \leq \left(\frac{dR_2(p)}{dp}\right)^{-2} \mathbb{E}\left[(\bar{Y}_2 - R_2(p))^2 | p \in J\right] \leq \frac{1}{N}\left(\frac{1}{C}\sqrt{\frac{p}{m}}\right)^2 = O\left(\frac{p}{mN}\right),$$

where we use Lemma A.9 and the fact that $R_2(p) \leq 1$.

When Case 2 holds, we can prove it similarly by inverting $R_3(p)$. When Case 3 holds, we have $p \leq 8 C_I/m$. Note that

$$R_4(p) = 1 - (1-p)^m \leq mp,$$

and then

$$\mathbb{E}\left[(\hat{p}-p)^2 \,|p \in J\right] \leq \left(\frac{dR_4(p)}{dp}\right)^{-2} \mathbb{E}\left[(\bar{Y}_4 - R_4(p))^2 | p \in J\right] \leq \frac{mp}{N} \cdot \frac{1}{C^2 m^2} = O\left(\frac{p}{mN}\right).$$

This concludes the proof of Theorem A.3. $\qquad\square$

It only remains to establish Lemmas A.8 and A.9, which we do next.

### A.2.1  Proof of Lemma A.8

For $p \in [0,1]$, consider the following interval

$$B_p(C) = \left[p - C\max\left\{\frac{1}{m}, \sqrt{\frac{p}{m}}\right\}, p + C\max\left\{\frac{1}{m}, \sqrt{\frac{p}{m}}\right\}\right] \cap [0,1].$$

Let $C = \sqrt{C_I}/25 = 32$, then $J \subseteq B_p(C)$.

We first prove the lemma for $p \leq 1/2$. $p \geq 1/2$ holds by symmetry (applying the same argument with $1 - p$). If $p \leq 64 C_I/m$, then

$$p + C\max\left\{\frac{1}{m}, \sqrt{\frac{p}{m}}\right\} \leq \frac{1}{m}(64 C_I + C\sqrt{64 C_I}) \leq \frac{65 C_I}{m}.$$

Hence $J \subseteq B_p(C) \subseteq [0, 65 C_I/m]$.

If $p > 64 C_I/m$, we fix some $i \geq 8$. Notice that

$$|I_i' \cap J_i'| \geq \frac{C_I i}{10m} - \frac{C_I}{2m}, \quad |I_{i+1}' \cap J_i'| \geq \frac{C_I i}{10m} + \frac{C_I}{20m}.$$

Furthermore, for $p \in I_i$,

$$C\sqrt{\frac{p}{m}} \leq C\frac{\sqrt{C_I}(i+1)}{m} = \frac{C_I(i+1)}{25m}.$$

If $p \in I_i' \cap J_i'$, then either $p + C\sqrt{\frac{p}{m}} \leq l_i + \frac{0.55C_Ii}{m}$, or $p - C\sqrt{\frac{p}{m}} \geq j_i - \frac{0.55C_Ii}{m}$. Otherwise we would have

$$2C\sqrt{\frac{p}{m}} \geq \frac{C_Ii}{10m} - \frac{C_I}{2m},$$

contradiction. If the former is true, then $J \subseteq B_p(C) \subseteq I_i'$. Else, $J \subseteq B_p(C) \subseteq J_i'$.

If $p \in I_{i+1}' \cap J_i'$, then similarly either $B_p(C) \subseteq I_{i+1}'$ or $B_p(C) \subseteq J_i'$.

Finally we consider the case when $p$ is not in any of the intersections above. If $p < j_i - \frac{0.55C_Ii}{m}$, then

$$p + C\sqrt{\frac{p}{m}} \leq \frac{1}{m}\left(C_Ii^2 + C_Ii + \frac{C_I}{2} - 0.55C_Ii + \frac{C_I(i+1)}{25}\right) \leq \frac{C_Ii^2}{m} + \frac{0.55C_Ii}{m}.$$

Hence $B_p(C) \subseteq I_i'$.

Similarly, if $p > j_i + \frac{0.55C_Ii}{m}$, we have $B_p(C) \subseteq I_{i+1}'$. If $p \in (l_i + \frac{0.55C_Ii}{m}, l_{i+1} - \frac{0.55C_Ii}{m})$, then $B_p(C) \subseteq J_i'$. $\qquad\square$

### A.2.2 Proof of Lemma A.9

First, for convenience we define a suitable random variable $S_m(p) \sim \text{Bin}(m, p)$. Let $t \in [0, m]$. Define the binomial tail,

$$P_m(p, t) = \Pr_{Z \sim \text{Bin}(m,p)}[Z \geq t] = \sum_{i \in [t,m]} \binom{m}{i} p^i (1-p)^{m-i}.$$

For an interval $I = [a, b)$, we define the probability mass of $\text{Bin}(m, p)$ inside $I$ as

$$P_m(p, I) := P_m(p, a) - P_m(p, b) = \sum_{i \in I} \binom{m}{i} p^i (1-p)^{m-i}.$$

Our argument will require the following results.

**Claim A.10.** *Let* $C_u{}^2/m \leq p \leq 1/2$ *for some* $C_u < 1/2$ *and* $t/m \in [p - C_u\sqrt{\frac{p}{m}}, p + C_u\sqrt{\frac{p}{m}}]$. *Then, for some constant* $C_0$,

$$\frac{\partial}{\partial p} P_m(p, t) \geq C_0 e^{-2C_u{}^2} \sqrt{\frac{m}{p}}.$$

*Proof.* Due to the binomial-beta relation [5, Equation 2.14], we have for positive integers $n$ and integers $t \in [n]$,

$$\Pr[S_m(p) \geq t] = n \int_0^p \Pr[S_{m-1}(u) = t - 1]du$$

Hence,

$$\frac{\partial}{\partial p} P_m(p, t) = m \Pr[S_{m-1}(p) = t - 1].$$

We use Stirling's approximation to bound the probability mass. For all integers $n$,

$$\sqrt{2\pi}n^{n+1/2}e^{-n} \leq n! \leq en^{n+1/2}e^{-n}.$$

We bound the probability mass of binomial:

$$\begin{aligned}
\Pr[S_m(p) = t] &= \binom{m}{t} p^t (1-p)^{m-t} \\
&\geq \frac{\sqrt{2\pi}}{e^2\sqrt{m}} \frac{1}{\sqrt{t/m}\sqrt{1-t/m}} \frac{p^t(1-p)^{m-t}}{(t/m)^t(1-t/m)^{m-t}} \\
&= \frac{\sqrt{2\pi}}{e^2\sqrt{m}} \frac{1}{\sqrt{t/m}\sqrt{1-t/m}} e^{-m\text{KL}(t/m||p)} \\
&\geq \frac{\sqrt{2\pi}}{e^2\sqrt{m}} \frac{1}{\sqrt{t/m}\sqrt{1-t/m}} e^{-m\frac{(t/m-p)^2}{p(1-p)}} \\
&\geq \frac{\sqrt{2\pi}}{e^2\sqrt{t}} e^{-m\frac{(t/m-p)^2}{p(1-p)}}
\end{aligned}$$

Hence for $t/m \in [p - C_u\sqrt{p/m}, p + C_u\sqrt{p/m}]$,

$$\Pr[S_{m-1}(p) = t - 1] \geq \frac{\sqrt{2\pi}}{e^2\sqrt{t-1}}e^{-\frac{C_u^2}{1-p}} \geq \frac{\sqrt{2\pi}}{e^2\sqrt{t-1}}e^{-2C_u^2} \geq \frac{\sqrt{2\pi}}{e^2\sqrt{2mp}}e^{-2C_u^2}$$

The final inequality is due to $p \geq C_u^2/m$, and hence $t \leq mp + C_u\sqrt{mp} \leq 2mp$. Multiplying the inequality by $m$ completes the proof. $\square$

**Claim A.11.** *Let $C_l$ be a constant, $(2C_l)^2/m \leq p \leq 1/2$, and $|p - t/m| > C_l\sqrt{\frac{p}{m}}$. Then there exists a constant $C_0'$ such that*

$$\frac{\partial}{\partial p}P_m(p,t) \leq C_0' e^{-C_l^2/3}\sqrt{\frac{m}{p}}.$$

*Proof.* We only need to prove the inequality for $t$ such that $|p - t/m| = C_l\sqrt{p/m}$. Suppppose that for $t/m = p + C_l\sqrt{p/m}$, we have

$$\frac{\partial}{\partial p}P_m(p,t) \leq C_0' e^{-C_l^2/3}\sqrt{\frac{m}{p}}.$$

Then for $t' > p + C_l\sqrt{p/m}$, since

$$P_m(p,t) = P_m(p,[t,t')) + P_m(p,t')$$

and

$$\frac{\partial P_m(p,[t,t'-1))}{\partial p} \geq 0, \frac{\partial P_m(p,t)}{\partial p} \geq 0, \frac{\partial P_m(p,t')}{\partial p} \geq 0,$$

we must have

$$\frac{\partial P_m(p,t')}{\partial p} \leq \frac{\partial P_m(p,t)}{\partial p} \leq C_0' e^{-C_l^2/3}\sqrt{\frac{m}{p}}.$$

A similar statement holds for $t/m = p - C_l\sqrt{p/m}$ and $t' < p - C_l\sqrt{p/m}$.

Hence we can assume that $|t/m - p| = C_l\sqrt{p/m}$. We again use the binomial-beta relation

$$\frac{\partial}{\partial p}P_m(p,t) = m\Pr[S_{m-1}(p) = t - 1].$$

We need to upper bound the probability mass.

$$\Pr[S_m(p) = t] = \binom{m}{t}p^t(1-p)^{m-t}$$

$$\leq \frac{e}{2\pi}\frac{1}{\sqrt{t/m}\sqrt{1-t/m}}\frac{p^t(1-p)^{m-t}}{(t/m)^t(1-t/m)^{m-t}}$$

$$= \frac{e}{2\pi\sqrt{m}}\frac{1}{\sqrt{t/m}\sqrt{1-t/m}}e^{-m\mathrm{KL}(t/m||p)}.$$

Since $p \geq (2C_l)^2/m$, we have $C_l\sqrt{p/m} \leq p/2$. Since $|t/m - p| = C_l\sqrt{p/m}$,

$$\frac{t}{m}\left(1 - \frac{t}{m}\right) \geq \min\left\{\left(p - C_l\sqrt{\frac{p}{m}}\right)\left(1 - p + C_l\sqrt{\frac{p}{m}}\right), \left(p + C_l\sqrt{\frac{p}{m}}\right)\left(1 - p - C_l\sqrt{\frac{p}{m}}\right)\right\}$$

$$\geq \frac{p}{4}.$$

Furthermore, since $\mathrm{KL}(p \,||\, q) \geq (p-q)^2/(2\max\{p,q\})$, when $t/m = C_l\sqrt{p/m}$ we have

$$\mathrm{KL}(t/m \,||\, p) \geq \frac{C_l^2 p}{2m(p + C_l\sqrt{p/m})} \geq \frac{C_l^2}{3m}.$$

The final inequality is due to $p \geq (2C_l)^2/m$ and hence $2C_l\sqrt{mp} \leq mp$. Combining the above results yields

$$\Pr(S_m(p) = t) \leq \frac{2e}{\pi\sqrt{mp}}e^{-C_l^2/3}.$$

$\square$

We now have the following claim, which establishes the first two items of Lemma A.9 in the regime $p \gg 1/m$.

**Claim A.12.** *Let $p > 64C_I/m$. Given fixed $i \leq r$, then in $I'_i$*

$$\left| \frac{dR_2(p)}{dp} \right| \geq C\sqrt{\frac{m}{p}}.$$

*A similar statement holds for $R_3$ inside $J'_i$.*

*Proof.* Recall that

$$I'_i = \left[ \frac{C_I i^2}{m} - \frac{0.55 C_I i}{m}, \frac{C_I i^2}{m} + \frac{0.55 C_I i}{m} \right].$$

Let $t = C_I i^2$, and note that for $p \in I'_i$

$$\left| p - \frac{t}{m} \right| \leq \frac{0.55 C_I i}{m} \leq 0.55\sqrt{C_I}\sqrt{\frac{p}{m}}.$$

Thus, letting $C_u = 0.55\sqrt{C_I}$, by Claim A.10

$$\frac{\partial P_m(p, t)}{\partial p} \geq C_0 e^{-2 \cdot (0.55)^2 C_I} \sqrt{\frac{m}{p}}.$$

For $t = (C_I j)^2$ where $|j - i| = 1$,

$$\left| p - \frac{t}{m} \right| \geq \frac{1.45 C_I i}{m} \geq 1.45\sqrt{C_I}\sqrt{\frac{p}{m}}.$$

Hence, letting $C_l = 1.45\sqrt{C_I}$, by Claim A.11

$$\frac{\partial P_m(p, t)}{\partial p} \leq C'_0 e^{-1.45^2 C_I/3} \sqrt{\frac{m}{p}}.$$

Thus for $\sqrt{C_I}/16 = 32$, we have $C_0 e^{-2 \cdot (0.55)^2 C_I/} \geq 3C'_0 e^{-1.45^2 C_I/3}$.

Note that

$$R_2(p) = \sum_j P_m(p, [ml_{2j}, ml_{2j+1}]) = \sum_j P_m(p, C_I(2j)^2) - P_m(p, C_I(2j+1)^2).$$

Also for $t' > t$,

$$\frac{\partial P_m(p, t)}{\partial p} \geq \frac{\partial P_m(p, t')}{\partial p}.$$

Therefore letting $C = C'_0 e^{-1.45^2 C_I/3}$, we have

$$\left| \frac{dR_2(p)}{dp} \right| \geq \left| \frac{\partial P_m(p, C_I i^2)}{\partial p} \right| - \left| \frac{\partial P_m(p, C_I(i-1)^2)}{\partial p} \right| - \left| \frac{\partial P_m(p, C_I(i+1)^2)}{\partial p} \right| \geq C\sqrt{\frac{m}{p}}$$

establishing the claim. $\square$

The above claim provides guarantees for $p > 64C_I/m$. It remains to prove the case for $p \leq 64C_I/m$.

**Claim A.13.** *For $p \leq 64C_I/m$, we have $\frac{dR_4(p)}{dp} \geq Cm$, for some constant $C > 0$. A similar statement holds for $R_2$ and $R_3$.*

*Proof.* The proof is straightforward since $R_4(p) = 1 - (1-p)^m$. For $p \leq 64C_I/m$, there must exist a constant $C' > 1$ such that

$$\frac{dR_4(p)}{dp} = m(1-p)^{m-1} \geq m\left(1 - \frac{C_I}{m}\right)^m \geq \frac{m}{C'} e^{-64C_I}. \qquad \square$$

Together, the two claims above establish the first two items of Lemma A.9, and the last claimm further shows the third item. This concludes the proof of Lemma A.9.

# B  Extended proof for the regime $m < k, \ell > \log(k/m)$

In this section, we provide the complete proof for the rate in Theorem 1.2 in the regime where $m < k, \ell > \log(k/m)$. We first recall the protocol.

Divide the domain $[k]$ into $t = 2m$ non-overlapping blocks $B_1, \ldots, B_t$, each with size $\lceil k/(2m) \rceil$. For $\mathbf{p} \in \Delta_k$, let $\mathbf{p}_B$ be the distribution over the blocks induced by $\mathbf{p}$, where for all $j \in [t]$ $\mathbf{p}_B(j) = \sum_{x \in B_j} \mathbf{p}_x$. For block $j \in [t]$, let $\bar{\mathbf{p}}_j(x)$ be the normalized distribution over elements in set $B_j$, i.e., for all $x \in B_j$, $\bar{\mathbf{p}}_j(x) = \mathbf{p}(x)/\mathbf{p}_B(j)$ (if $\mathbf{p}_B(j) = 0$, we set $\bar{\mathbf{p}}_j(x) = 1/(2^\ell - 1)$), which is the uniform distribution over $B_j$). The protocol then estimates $\mathbf{p}_B$ and $\bar{\mathbf{p}}_j$'s separately.

Without loss of generality, we assume each user has $\ell \geq 2\log(k/m)$ bits, else each user can just use their first $\log(k/m)$ bits to conduct the protocol for the regime $m2^\ell \leq k$ as described in Section 3.1 and obtain the same rate up to constants. Letting $\ell_0 := \ell/2$, the protocol is as follows.

**Estimating $\mathbf{p}_B$.** Each user maps the observed samples to the set they belong to in $B_1, \ldots, B_t$. Then users use the first $\ell_0$ bits to conduct the protocol described in Section 3.3, i.e. the protocol for the regime $m = \Theta(k)$[6], to obtain an estimate $\widehat{\mathbf{p}}_B$. This is feasible since the domain size of $\mathbf{p}_B$ is $t = 2m = \Theta(m)$.

**Estimating normalized distributions.** Since each block is only of size $\lceil k/(2m) \rceil$, given a block index, each user can send an element within the block with $\log \lceil k/(2m) \rceil < \ell_0$ bits. To take advantage of this, we assign each user $t' = \lfloor \ell_0 / \log \lceil k/(2m) \rceil \rfloor$ different blocks. More precisely, user $i$ is assigned blocks $B_j$ for $j = (i-1)t' + 1, (i-1)t' + 2, \ldots, it' \mod t$. For each assigned $B_j$, the user sends the first sample they observe in $B_j$ or sends $\perp$ if none of them appears.

In total, $n$ users send out $nt'$ messages, which we index as $(Z_i)_{i \in [nt']}$. Based on which of the $t$ blocks the messages correspond to, we can divide them into $t$ sets where $S_j = \{i \in [n't] \mid j = i \mod t\}$. The server collects the messages from the users and uses the empirical estimator to estimate each normalized distribution within each block, where for $j \in [2t], x \in [\lceil k/(2m) \rceil]$,

$$\widehat{\mathbf{p}}_j(x) = \frac{\sum_{i \in S_j} \mathbb{1}\{Z_i = x\}}{\sum_{i \in S_j} \mathbb{1}\{Z_i \neq x\}}. \tag{9}$$

Accordingly, the final estimates are

$$\widehat{\mathbf{p}}(x) = \widehat{\mathbf{p}}_B(j) \cdot \widehat{\mathbf{p}}_j(x), \qquad \forall j \in [t], x \in B_j. \tag{10}$$

**Error for estimating $\mathbf{p}_B$.** Using the bound we obtained in Section 3.3 (the second upper bound in Theorem 1.2), and using the fact that the domain size is $t$ for $\mathbf{p}_B$, we get

$$\mathbb{E}[\text{TV}(\widehat{\mathbf{p}}_B, \mathbf{p}_B)] = O\left(\sqrt{\frac{t^2}{mn\ell_0}}\right) = O\left(\sqrt{\frac{m}{n\ell}}\right). \tag{11}$$

**Error for estimating $\bar{\mathbf{p}}_j$'s.** Recall the following two facts about the messages.

1. For $j \in [t]$, let $N_j := \sum_{i \in S_j} \mathbb{1}\{Z_i \neq \perp\}$, the number of samples the server receives from $B_j$. Then $N_j$ follows a binomial distribution $\text{Bin}(\lfloor n/t \rfloor, \beta_j)$ where $\beta_j := 1 - (1 - \mathbf{p}_B(j))^m = \Theta(\min\{m\mathbf{p}_B(j), 1\})$.
2. For $j \in [t]$, conditioned on the fact that the first sample in $B_j$ is sent, the sample is distributed according to the normalized distribution $\bar{\mathbf{p}}_j$.

Similar to Lemma 3.2 in the main text, we have the following lemma.

**Lemma B.1.** *If $\mathbf{p}_B(j) > 0$, $\mathbb{E}[\text{TV}(\widehat{\mathbf{p}}_j, \bar{\mathbf{p}}_j)] = O\left(\sqrt{\frac{k}{m|S_j|\beta_j}}\right)$.*

*Proof.* By a Chernoff bound, we have $\Pr\left(N_j < \frac{|S_j|\beta_j}{2}\right) \leq \exp\left(-\frac{|S_j|\beta_j}{8}\right)$. When $N_j \geq \frac{|S_j|\beta_j}{2}$, it is folklore that

$$\mathbb{E}\left[\text{TV}(\widehat{\mathbf{p}}_j, \bar{\mathbf{p}}_j) \mid N_j \geq \frac{|S_j|\beta_j}{2}\right] = O\left(\sqrt{\frac{\lceil k/2m \rceil}{N_j}}\right) = O\left(\sqrt{\frac{k}{m|S_j|\beta_j}}\right).$$

---

[6]The protocol in Section 3.3 requires $m > k$, but the same protocol applies for the case when $m = \Theta(k)$ with the same guarantee.

Combining both, we get

$$\mathbb{E}[\mathrm{TV}(\widehat{\mathbf{p}}_j, \bar{\mathbf{p}}_j)] = O\left(\sqrt{\tfrac{k}{m|S_j|\beta_j}}\right) + \exp\left(-\tfrac{|S_j|\beta_j}{8}\right) = O\left(\sqrt{\tfrac{k}{m|S_j|\beta_j}}\right),$$

completing the proof of the lemma. □

By the way the blocks are assigned to users, it can be seen that for all $j \in [k]$,

$$|S_j| \geq \left\lfloor \frac{nt'}{t} \right\rfloor = \Theta\left(\frac{n\ell}{m\log(k/m+1)}\right).$$

Hence combining with Lemma B.1, we have if $\mathbf{p}_B(j) > 0$,

$$\mathbb{E}[\mathrm{TV}(\widehat{\mathbf{p}}_j, \bar{\mathbf{p}}_j)] = O\left(\sqrt{\frac{k\log(k/m+1)}{n\ell\beta_j}}\right). \tag{12}$$

Recall in Lemma 3.1, we have

$$\mathbb{E}[\mathrm{TV}(\widehat{\mathbf{p}}, \mathbf{p})] \leq \mathbb{E}[\mathrm{TV}(\widehat{\mathbf{p}}_B, \mathbf{p}_B)] + \sum_{j\in[t]} \mathbf{p}_B(j)\mathbb{E}[\mathrm{TV}(\widehat{\mathbf{p}}_j, \bar{\mathbf{p}}_j)].$$

Pluggin in Eq. (11) and Eq. (12), we have

$$
\begin{aligned}
\mathbb{E}[\mathrm{TV}(\widehat{\mathbf{p}}, \mathbf{p})] &= O\left(\sqrt{\tfrac{m}{n\ell}} + \sum_{j\in[t]} \mathbf{p}_B(j)\sqrt{\frac{k\log(k/m+1)}{n\ell\beta_j}}\right) \\
&= O\left(\sqrt{\tfrac{m}{n\ell}} + \sum_{j\in[t]}\left(\mathbf{p}_B(j)\sqrt{\frac{k\log(k/m+1)}{n\ell}} + \sqrt{\frac{\mathbf{p}_B(j)k\log(k/m+1)}{mn\ell}}\right)\right) \\
&= O\left(\sqrt{\tfrac{m}{n\ell}} + \sqrt{\frac{k\log(k/m+1)}{n\ell}}\right) \\
&= O\left(\sqrt{\frac{k\log(k/m+1)}{n\ell}}\right), \tag{13}
\end{aligned}
$$

where the last inequality is due to $m < k$.

## C   Lower bounds

In this section, we provide self-contained proofs of the three lower bounds: the lower bound of Theorem 1.1 for $m \leq k/2^\ell$, and the two distinct lower bounds of Theorem 1.3 (under the assumption that $n > (k/\ell)^2$), for $k/2^\ell < m \leq k\log k$ and $m > k\log k$, respectively. The proof follows the outline sketched in Section 4 with a slight tweak in the information measure we bound for better presentation. The same argument holds for the information measure used in Section 4 as well.

We begin with a simple lemma, which shows that in order to prove minimax rate lower bounds in the multinomial setting (where each user receives exactly $m$ i.i.d. samples from the unknown distribution $\mathbf{p}$) it is enough to establish lower bounds in the Poisson setting (where each user receives $M_t \sim \mathrm{Poi}(m)$ samples, the $M_t$'s being drawn independently). This simple fact will facilitate some of our arguments, as after this "Poissonization" the numbers of occurrences of each domain element will be independent across both users and domain elements, with the number of occurrence of $i \in [k]$ at each user following a $\mathrm{Poi}(m\mathbf{p}_i)$ distribution.

Formally, let the multinomial and Poissonized settings be defined as follows:

**MULTINOMIAL**$(n, m)$**:** each of the $n$ users obtains $m$ samples from $\mathbf{p}$. The $mn$ samples are i.i.d.

**POISSONIZED**$(n, m)$**:** For $1 \leq t \leq n$, user $t$ observing $M_t$ samples from $\mathbf{p}$, where $(M_t)_{1\leq t\leq n}$ are independent $\mathrm{Poi}(m)$. The $\sum_{t=1}^n M_t$ samples are i.i.d.

**Lemma C.1** (Reduction to the Poissonized Model). *Suppose that there exists an $\ell$-bit protocol for estimation in the* MULTINOMIAL$(n, m)$ *model, with expected error rate $\varepsilon$. Then there exists an $(\ell + 1)$-bit protocol for estimation in the* POISSONIZED$(20n, 2m)$ *model, with expected error rate $\varepsilon + e^{-2n/3}$. Moreover, the latter protocol is noninteractive if the former one was.*

Given the information-theoretic bound showing that having $n\ell \gtrsim k \log(1/\varepsilon)$ is required (from a packing argument), the $e^{-2n/3}$ term can for instance be ignored whenever $\ell \lesssim k$. Overall, the above lemma shows that, up to constant factors in $m$ and $n$, establishing a rate lower bound in the Poissonized model implies the same lower bound in the multinomial model.

*Proof.* The lemma follows from standard concentration of Poisson random variables. The probability that a $\mathrm{Poi}(2m)$ random variable $M$ is less than $m$ is bounded as

$$\Pr[\, M < \mathbb{E}[M]/2 \,] \leq e^{-\frac{m}{6}} \leq e^{-\frac{1}{6}}$$

and thus each of the $20n$ users gets at least $m$ samples independently with probability at least $1 - e^{-\frac{1}{6}} > 3/20$. By a Chernoff bound, this means that at least $n$ (out of the $20n$) users will obtain at least $m$ samples, except with probability at most $e^{-\frac{2}{3}n}$.

The $20n$ users can then use the extra bit of communication to indicate whether they indeed received at least $m$ samples, and the remaining $\ell$ to follow the MULTINOMIAL$(n, m)$ $\ell$-bit protocol. Of course, if fewer than $n$ users got at least $m$ samples, they will not be able to fully simulate that protocol, but given above bound on the probability $p$ this happens, we can bound the resulting expected error as $\varepsilon \cdot (1 - p) + 1 \cdot p \leq \varepsilon + e^{-2n/3}$, as claimed. $\qquad\square$

In view of this lemma, we hereafter focus on establishing our two lower bounds in the POISSONIZED$(n, m)$ model; that is, to lower bound the following quantity

$$\mathcal{R}^{\mathrm{Poi}}(\ell, k, n, m) := \min_{W^n \in \mathcal{W}_\ell^n} \min_{\hat{\mathbf{p}}} \max_{\mathbf{p} \in \Delta_k} \mathbb{E}[\mathrm{TV}(\hat{\mathbf{p}}(Y^n), \mathbf{p})] \tag{14}$$

where the minimum is taken over all (possibly interactive) protocols in the POISSONIZED$(n, m)$ setting.

## C.1 Preliminaries: The lower bound framework of [4]

This section summarizes the lower bound framework of Acharya et al. [4], which will be a key ingredient in our lower proofs. Let $\mathcal{Z} := \{-1, +1\}^k$ and $\{\mathbf{p}_z\}_{z \in \mathcal{Z}}$ (where $\mathbf{p}_z = \mathbf{p}_{\theta_z}$) be a collection of distributions over $\mathcal{X}$, indexed by $z \in \mathcal{Z}$. For $z \in \mathcal{Z}$, denote by $z^{\oplus i} \in \mathcal{Z}$ the vector obtained by flipping the sign of the $i$th coordinate of $z$.

**Assumption 1.** For every $z \in \mathcal{Z}$ and $i \in [k]$ it holds that $\mathbf{p}_{z^{\oplus i}} \ll \mathbf{p}_z$, and there exist measurable functions $\phi_{z,i} \colon \mathcal{X} \to \mathbb{R}$ such that

$$\frac{d\mathbf{p}_{z^{\oplus i}}}{d\mathbf{p}_z} = 1 + \alpha_{z,i}\phi_{z,i},$$

where $|\alpha_{z,i}| \leq \alpha$ for some constant $\alpha \in \mathbb{R}$ independent of $z, i$.

**Assumption 2.** There exists some $\kappa_\mathcal{W} \in [1, \infty]$ such that

$$\max_{z \in \mathcal{Z}, y \in \mathcal{Y}} \sup_{W \in \mathcal{W}} \frac{\mathbb{E}_{\mathbf{p}_{z^{\oplus i}}}[W(y \mid X)]}{\mathbb{E}_{\mathbf{p}_z}[W(y \mid X)]} \leq \kappa_\mathcal{W}. \tag{15}$$

**Assumption 3.** For all $z \in \mathcal{Z}$ and $i, j \in [k]$, $\mathbb{E}_{\mathbf{p}_z}[\phi_{z,i}\phi_{z,j}] = \mathbb{1}\{i = j\}$. (In particular, $\mathbb{E}_{\mathbf{p}_z}[\phi_{z,i}^2] = 1$.)

**Assumption 4.** There exists some $\sigma \geq 0$ such that, for all $z \in \mathcal{Z}$, the random vector $\phi_z(X) := (\phi_{z,i}(X))_{i \in [k]} \in \mathbb{R}^k$ is $\sigma^2$-subgaussian for $X \sim \mathbf{p}_z$, with independent coordinates.

Let $Z$ be a random variable over $\mathcal{Z}$ with i.i.d. coordinates with parameter $\theta \in (0, 1/2)$ (*i.e.*, $\mathbb{E}[Z_i] = 2\theta - 1$ for all $i \in [k]$). Then the following holds:

**Theorem C.2** (Main theorem of [4]). *Fix $\theta \in (0, 1/2]$. Let $\Pi$ be a sequentially interactive protocol using $\mathcal{W}$, and let $Z$ be a random variable on $\mathcal{Z}$ with distribution $\mathrm{Rad}(\theta^{\otimes k})$. Let $(Y^n, U)$ be the transcript of $\Pi$ when the input $X_1, \ldots, X_n$ is i.i.d. with common distribution $\mathbf{p}_Z$. Then, under Assumption 1, 2, we have*

$$\left( \frac{1}{k} \sum_{i=1}^{k} \mathrm{TV}\left( \mathbf{p}_{+i}^{Y^n}, \mathbf{p}_{-i}^{Y^n} \right) \right)^2 \leq \frac{2}{k} \left( \kappa_{\mathcal{W}} \wedge \theta^{-1} \right) n\alpha^2 \max_{z \in \mathcal{Z}} \max_{W \in \mathcal{W}} \sum_{i=1}^{k} \sum_{y \in \mathcal{Y}} \frac{\mathbb{E}_{\mathbf{p}_z}[\phi_{z,i}(X)W(y \mid X)]^2}{\mathbb{E}_{\mathbf{p}_z}[W(y \mid X)]},$$

(16)

*where $\mathbf{p}_{+i}^{Y^n} := \mathbb{E}\left[ \mathbf{p}_Z^{Y^n} \mid Z_i = 1 \right]$, $\mathbf{p}_{-i}^{Y^n} := \mathbb{E}\left[ \mathbf{p}_Z^{Y^n} \mid Z_i = 1 \right]$. Moreover, under the additional Assumption 3,*

$$\left( \frac{1}{k} \sum_{i=1}^{k} \mathrm{TV}\left( \mathbf{p}_{+i}^{Y^n}, \mathbf{p}_{-i}^{Y^n} \right) \right)^2 \leq \frac{2}{k} \left( \kappa_{\mathcal{W}} \wedge \theta^{-1} \right) n\alpha^2 \max_{z \in \mathcal{Z}} \max_{W \in \mathcal{W}} \sum_{y \in \mathcal{Y}} \frac{\mathrm{Var}_{\mathbf{p}_z}[W(y \mid X)]}{\mathbb{E}_{\mathbf{p}_z}[W(y \mid X)]},$$

(17)

*Finally, if Assumption 4 holds as well, we have*

$$\left( \frac{1}{k} \sum_{i=1}^{k} \mathrm{TV}\left( \mathbf{p}_{+i}^{Y^n}, \mathbf{p}_{-i}^{Y^n} \right) \right)^2 \leq \frac{\ln 2}{k} \left( \kappa_{\mathcal{W}} \wedge \theta^{-1} \right) \cdot n\alpha^2 \sigma^2 \max_{z \in \mathcal{Z}} \max_{W \in \mathcal{W}} I(\mathbf{p}_z; W),$$

(18)

*where $I(\mathbf{p}_z; W)$ denotes the mutual information $I(X; Y)$ between the input $X \sim \mathbf{p}_z$ and the output $Y$ of the channel $W$ with $X$ as input.*

Note that, as in [4], one can bound the quantity $\sum_{y \in \mathcal{Y}} \frac{\mathrm{Var}_{\mathbf{p}_z}[W(y|X)]}{\mathbb{E}_{\mathbf{p}_z}[W(y|X)]}$ from the right-hand-side of Eq. (17) by $|\mathcal{Y}| = 2^\ell$, and the quantity $I(\mathbf{p}_z; W)$ from the right-hand-side of Eq. (18) by $\log |\mathcal{Y}| \lesssim \ell$.

In Section 4, $\sum_{i \in [k]} I(Z_i; Y^n)$ is used as the information measure to bound. When $Z_i \sim \mathrm{Bern}(1/2)$, $I(Z_i; Y^n)$ and $\mathrm{TV}\left( \mathbf{p}_{+i}^{Y^n}, \mathbf{p}_{-i}^{Y^n} \right)^2$ are within constant factor of each other. In this proof, we focus on deriving upper and lower bounds on $\sum_{i \in [k]} \mathrm{TV}\left( \mathbf{p}_{+i}^{Y^n}, \mathbf{p}_{-i}^{Y^n} \right)$ for presentation purposes. The same bound applies for $\sum_{i \in [k]} I(Z_i; Y^n)$ up to constant factors using the proof in [4].

## C.2   Our lower bound instances

In order to apply the results from Appendix C.1, we need to describe the family of distributions $\{\mathbf{p}_z\}_{z \in \{-1,+1\}^k}$ we consider, and show that it satisfies the assumptions with some values of $\alpha$, $\kappa$, and $\theta$. Let $\gamma \in (0, 1/2)$ be the purported expected error rate, and define, for $z \in \{-1, +1\}^k$, $\mathbf{p}_z$ as the probability distribution over $\mathbb{N}^k$ such that

$$\mathbf{p}_z = \bigotimes_{i=1}^{k} \mathrm{Poi}(\theta_z(i))$$

(19)

where

$$\theta_z(i) = \frac{m(1 + 2\gamma z_i)}{k}, \qquad i \in [k].$$

(20)

We will choose the distribution of $Z$ to be uniform over $\{-1, +1\}^k$; that is, $Z \sim \mathrm{Rad}(\theta^{\otimes k})$ for $\theta = 1/2$. With this choice, we get the following:

**Lemma C.3.** *The family $\{\mathbf{p}_z\}_{z \in \{-1,+1\}^k}$ of probability distributions over $\mathbb{N}^k$ satisfies Assumption 1, 3, and 2 with*

$$\alpha := \sqrt{e^{8m\gamma^2/k} - 1} = O\left( \sqrt{m\gamma^2/k} \right),$$

*where the asymptotics are as $\gamma \to 0$; and, for $i \in [k]$ and $z \in \{-1, +1\}^k$,*

$$\alpha_{z,i} := \sqrt{e^{\frac{4m\gamma^2}{k(1+\gamma z_i)}} - 1}, \qquad \phi_{z,i}(\mathbf{m}) := \frac{1}{\alpha_{z,i}} \left( \left( \frac{1 - \gamma z_i}{1 + \gamma z_i} \right)^{\mathbf{m}_i} e^{\frac{2m\gamma z_i}{k}} - 1 \right)$$

*and $\kappa = \infty$.*

*Proof.* The claimed value of $\kappa$ is trivially satisfied, so we only have to prove that our stated settings of $\alpha, \alpha_{z,i}$, and $\phi_{z,i}$ satisfy Assumption 1 and 3. Fix any $z \in \{-1, +1\}^k$. For any $\mathbf{m} = (\mathbf{m}_1, \ldots, \mathbf{m}_k) \in \mathbb{N}^k$, we have

$$\mathbf{p}_z(\mathbf{m}) = \prod_{i=1}^{k} \frac{\left(\frac{1+\gamma z_i}{k}\right)^{\mathbf{m}_i}}{\mathbf{m}_i!} e^{-\frac{m(1+\gamma z_i)}{k}}$$

and therefore, for every $i \in [k]$,

$$\frac{\mathbf{p}_{z \oplus i}(\mathbf{m})}{\mathbf{p}_z(\mathbf{m})} = \left(\frac{1-\gamma z_i}{1+\gamma z_i}\right)^{\mathbf{m}_i} e^{\frac{2m\gamma z_i}{k}}.$$

It can be verified, using the expression of the moment-generating function (MGF) of a Poisson distribution, that

$$\mathbb{E}_{\mathbf{p}_z}\left[\left(\frac{\mathbf{p}_{z \oplus i}(\mathbf{m})}{\mathbf{p}_z(\mathbf{m})} - 1\right)^2\right] = \exp\left(\frac{4m\gamma^2}{k} \frac{1}{1+\gamma z_i}\right) - 1$$

Letting $\alpha_{z,i} := \sqrt{\exp\left(\frac{4m\gamma^2}{k} \frac{1}{1+\gamma z_i}\right) - 1} \leq \sqrt{\exp\left(\frac{8m\gamma^2}{k}\right) - 1} := \alpha$ (since $\gamma \leq 1/2$), we thus have $\frac{\mathbf{p}_{z \oplus i}}{\mathbf{p}_z} = 1 + \alpha_{z,i}\phi_{z,i}$ where

$$\phi_{z,i}(\mathbf{m}) := \frac{1}{\alpha_{z,i}}\left(\left(\frac{1-\gamma z_i}{1+\gamma z_i}\right)^{\mathbf{m}_i} e^{\frac{2m\gamma z_i}{k}} - 1\right)$$

satisfies $\mathbb{E}_{\mathbf{p}_z}[\phi_{z,i}\phi_{z,j}] = \mathbb{1}\{i = j\}$ (by the previous computation for $i = j$, and using independence and $\mathbb{E}_{\mathbf{p}_z}\left[\frac{\mathbf{p}_{z \oplus i}(\mathbf{m})}{\mathbf{p}_z(\mathbf{m})}\right] = \mathbb{E}_{\mathbf{p}_z^{\oplus i}}[1] = 1$ for $i \neq j$). $\qquad\square$

*Remark* C.4 (Choice of prior for $Z$). We observe that our choice of prior for $Z$ (the uniform distribution on $\{-1, +1\}^k$) implies that the parameters $\theta_Z(i)$ (for $i \in [k]$) as defined in (20) may not sum to $m$:

$$1 - 2\gamma \leq \frac{1}{m}\sum_{i=1}^{k}\theta_Z(i) \leq 1 + 2\gamma, \qquad \mathbb{E}_Z\left[\frac{1}{m}\sum_{i=1}^{k}\theta_Z(i)\right] = 1$$

Given the correspondence to the univariate probability distribution case (for which a lower bound in our end goal), this may seem problematic, as of course a probability distribution is required to sum to one. However, due to the tight concentration of $\sum_{i=1}^{k}\theta_Z(i)$ around its mean and by standard renormalization arguments, the fact that our family of instances only corresponds to *approximate* probability distributions over $[k]$ does not affect the scope of the resulting lower bound.

Finally, in view of applying Theorem C.2 to derive a lower bound, we will require the following, relatively standard "Assouad-type bound":

**Lemma C.5.** *Let $\Pi$ be a sequentially interactive protocol for estimation achieving an expected minimax rate $\gamma$, and let $Z$ be a random variable uniformly distributed on $\{-1, +1\}^k$. Let $(Y^n, U)$ be the transcript of $\Pi$ when the input $X_1, \ldots, X_n$ is i.i.d. with common distribution $\mathbf{p}_Z$, where $\{\mathbf{p}_z\}_{z \in \{-1, +1\}^k}$ is as in (19). Then*

$$\sum_{i=1}^{k} \mathrm{TV}\left(\mathbf{p}_{+i}^{Y^n}, \mathbf{p}_{-i}^{Y^n}\right) = \Omega(k) \tag{21}$$

*where $\mathbf{p}_{+i}^{Y^n} := \mathbb{E}\left[\mathbf{p}_Z^{Y^n} \mid Z_i = 1\right]$, $\mathbf{p}_{-i}^{Y^n} := \mathbb{E}\left[\mathbf{p}_Z^{Y^n} \mid Z_i = 1\right]$.*

*Proof.* By an argument analogous to [4, Lemma 9], one can extract from the output $\hat{\mathbf{p}}$ of $\Pi$ an estimator $\hat{Z}$ of $Z$ such that, on the one hand,

$$\frac{1}{k}\sum_{i=1}^{k}\Pr\left(\hat{Z}_i \neq Z_i\right) \leq \frac{1}{10}.$$

On the other hand, considering for fixed $i \in [k]$ the Markov chain $Z_i \to Y^n \to \hat{Z}_i$, we get, by considering the hypothesis testing problem of distinguishing $Z_i = 1$ and $Z_i = -1$ given our uniform prior, that

$$\Pr\left(\hat{Z}_i \neq Z_i\right) \geq \frac{1}{2}\left(1 - \mathrm{TV}\left(\mathbf{p}_{+i}^{Y^n}, \mathbf{p}_{-i}^{Y^n}\right)\right).$$

Summing over $i \in [k]$ and combining the two bounds yields the result. $\square$

## C.3 The lower bound for $m < k/2^\ell$

We now prove our first lower bound:

**Theorem C.6.** *For all $m, n, \ell \geq 1$, the minimax rate in the Poissonized setting satisfies*

$$\mathcal{R}^{\mathrm{Poi}}(\ell, k, n, m) = \Omega\left(\sqrt{\frac{k}{mn}} \vee \sqrt{\frac{k^2}{mn2^\ell}}\right).$$

*Proof.* With Lemmas C.3 and C.5 in hand, the claimed rate lower bound is straightforward. First, note that by Lemma C.5, we have

$$\left(\frac{1}{k} \sum_{i=1}^{k} \mathrm{TV}\left(\mathbf{p}_{+i}^{Y^n}, \mathbf{p}_{-i}^{Y^n}\right)\right)^2 = \Omega(k).$$

By Lemma C.3 and Theorem C.2 (since $\theta = 1/2$, as our prior for $Z$ is uniform), we have

$$\left(\frac{1}{k} \sum_{i=1}^{k} \mathrm{TV}\left(\mathbf{p}_{+i}^{Y^n}, \mathbf{p}_{-i}^{Y^n}\right)\right)^2 \leq \frac{4}{k} n\alpha^2 \max_{z \in \mathcal{Z}} \max_{W \in \mathcal{W}} \sum_{y \in \mathcal{Y}} \frac{\mathrm{Var}_{\mathbf{p}_z}[W(y \mid X)]}{\mathbb{E}_{\mathbf{p}_z}[W(y \mid X)]} \leq \frac{4n\alpha^2}{k} 2^\ell = O\left(\frac{nm2^\ell \gamma^2}{k^2}\right).$$

Combining the two yields our claimed minimax lower bound on the rate $\gamma$:

$$\gamma = \Omega\left(\sqrt{\frac{k^2}{nm2^\ell}}\right).$$

The second part of the lower bound, $\gamma = \Omega\left(\sqrt{\frac{k}{nm}}\right)$, immediately follows from the known centralized minimax lower bound. $\square$

## C.4 The lower bound for $m \geq k \log k$ and $n > (k/\ell)^2$

We now prove our second lower bound:

**Theorem C.7.** *For all $m, n, \ell \geq 1$ such that $m \geq k \log k$ and $n > (k/\ell)^2$, the minimax rate in the Poissonized setting satisfies*

$$\mathcal{R}^{\mathrm{Poi}}(\ell, k, n, m) = \Omega\left(\sqrt{\frac{k}{mn}} \vee \sqrt{\frac{k^2}{mn\ell}}\right).$$

*Proof.* As before, the $\Omega\left(\sqrt{\frac{k}{mn}}\right)$ lower bound follows from the known bound in the centralized case, so it suffices to establish the second term.

In view of Theorem C.2, our goal would be to show that Assumption 4 is satisfied in this parameter regime, in order to obtain an $\ell$ dependence via the bounds provided by (18). Unfortunately, the functions $\phi_{z,i}$ obtained for our family $\{\mathbf{p}_z\}_{z \in \{-1,+1\}^k}$ (as per Lemma C.3) do *not* satisfy the desired subgaussian property. Still, we can get around this by decomposing the $\phi_{z,i}$ into several terms, among which one satisfies (18) and the others can be bounded through other means.

Io implement this roadmap, we first decompose $\phi_{z,i}$ into a *linear* and a *residual* term,

$$\phi_{z,i}(\mathbf{m}) = \frac{-2\gamma z_i}{\alpha_{z,i}}\left(\mathbf{m}_i - \frac{m(1 + \gamma z_i)}{k}\right) + \psi_{z,i}(\mathbf{m}), \tag{22}$$

where

$$\psi_{z,i}(\mathbf{m}) := \frac{1}{\alpha_{z,i}}\left(\left(\frac{1-\gamma z_i}{1+\gamma z_i}\right)^{\mathbf{m}_i} e^{\frac{2m\gamma z_i}{k}} - 1 + 2\gamma z_i\left(\mathbf{m}_i - \frac{m(1+\gamma z_i)}{k}\right)\right).$$

It can be shown, via computations involving the MGF of a Poisson distribution, that the residual term satisfies

$$\mathbb{E}_{\mathbf{p}_z}\left[\psi_{z,i}^2\right] = \frac{1}{\alpha_{z,i}^2}\left(\exp\left(\frac{4m\gamma^2}{k}\frac{1}{1+\gamma z_i}\right) - 1 - \frac{4m\gamma^2(1-\gamma z_i)}{k}\right) = O\left(\left(\frac{m}{k}+1\right)\gamma^2\right).$$

We then further decompose the linear term into a subgaussian term and another residual term, to account with the two regimes of tail behaviour of a Poisson random variable. Specifically, let

$$\zeta_{z,i}(\mathbf{m}) = \frac{-2\gamma z_i}{\alpha_{z,i}}\left(\mathbf{m}_i - \frac{m(1+\gamma z_i)}{k}\right)\mathbb{1}\{\mathbf{m}_i \le 10\theta_z(i)\} \tag{23}$$

$$\xi_{z,i}(\mathbf{m}) = \frac{-2\gamma z_i}{\alpha_{z,i}}\left(\mathbf{m}_i - \frac{m(1+\gamma z_i)}{k}\right)\mathbb{1}\{\mathbf{m}_i > 10\theta_z(i)\} \tag{24}$$

The next claim shows that, indeed, $\zeta_{z,i}(\mathbf{m})$ will now be subgaussian.

**Claim C.8.** *Fix any $i \in [k]$ and $z \in \{-1,+1\}^k$. For $\zeta_{z,i}$ defined in* (23)*, $\zeta_{z,i}(\mathbf{m})$ is $C$-subgaussian as $\mathbf{m} \sim \mathbf{p}_z$, $\zeta_{z,i}(\mathbf{m})$, where $C > 0$ is an absolute constant.*

*Proof.* Recall that $\mathbf{m}_i \sim \mathrm{Poi}(\theta_z(i))$ with $\theta_z(i) = \frac{m(1+\gamma z_i)}{k}$. Thus, for any $t \ge 0$,

$$\Pr\left(|\zeta_{z,i}(\mathbf{m})| \ge t\right) = \Pr\left(\zeta_{z,i}(\mathbf{m}) \le -t\right) + \Pr\left(\zeta_{z,i}(\mathbf{m}) \ge t\right)$$

$$= \Pr\left(\mathbf{m}_i \le \mathbb{E}[\mathbf{m}_i] - \frac{\alpha_{z,i}t}{2\gamma}\right) + \Pr\left((\mathbf{m}_i - \mathbb{E}[\mathbf{m}_i])\mathbb{1}\{\mathbf{m}_i \le 10\theta_z(i)\} \ge \frac{\alpha_{z,i}t}{2\gamma}\right)$$

By standard Poisson concentration bounds (*e.g.*, Bennett's inequality, or [9]), we have

$$\Pr\left(\mathbf{m}_i \le \mathbb{E}[\mathbf{m}_i] - \frac{\alpha_{z,i}t}{2\gamma}\right) \le \exp\left(-\frac{\alpha_{z,i}^2 t^2}{16\gamma^2\mathbb{E}[\mathbf{m}_i]}\right) \le e^{-t^2/16}, \tag{25}$$

since $\alpha_{z,i}^2 \ge \frac{2m\gamma^2}{k}$ and $\mathbb{E}[\mathbf{m}_i] \le \frac{2m}{k}$. For the other term, observing that

$$\Pr\left((\mathbf{m}_i - \mathbb{E}[\mathbf{m}_i])\mathbb{1}\{\mathbf{m}_i \le 10\mathbb{E}[\mathbf{m}_i]\} \ge u\right) = 0$$

whenever $u > 9\mathbb{E}[\mathbf{m}_i]$, we can focus on the case $t' := \frac{\alpha_{z,i}t}{2\gamma} \le 9\mathbb{E}[\mathbf{m}_i]$ and get

$$\Pr\left((\mathbf{m}_i - \mathbb{E}[\mathbf{m}_i])\mathbb{1}\{\mathbf{m}_i \le 10\mathbb{E}[\mathbf{m}_i]\} \ge \frac{\alpha_{z,i}t}{2\gamma}\right) \le \Pr\left((\mathbf{m}_i - \mathbb{E}[\mathbf{m}_i]) \ge t'\right)$$

$$\le \exp\left(-\frac{t'^2}{2(\mathbb{E}[\mathbf{m}_i] + t')}\right)$$

$$\le \exp\left(-\frac{t'^2}{20\mathbb{E}[\mathbf{m}_i]}\right)$$

$$\le \exp\left(-\frac{t^2}{80}\right)$$

using the same bounds $\alpha_{z,i}^2 \ge \frac{2m\gamma^2}{k}$ and $\mathbb{E}[\mathbf{m}_i] \le \frac{2m}{k}$. Overall, we have

$$\Pr\left(|\zeta_{z,i}(\mathbf{m})| \ge t\right) \le 2e^{-t^2/80}$$

for all $t \ge 0$, showing that $\zeta_{z,i}(\mathbf{m})$ is $C$-subgaussian for some absolute constant $C > 0$. $\square$

The second residual term, while not subgaussian, can be bounded directly through other means:

**Claim C.9.** *Fix any $i \in [k]$ and $z \in \{-1,+1\}^k$. For $\xi_{z,i}$ defined in* (24)*, we have $\mathbb{E}_{\mathbf{p}_z}\left[\xi_{z,i}^2\right] \le 8e^{-\frac{2m}{k}}$.*

*Proof.* Fix $i \in [k]$ and $z \in \{-1, +1\}^k$. By Cauchy–Schwarz,

$$\mathbb{E}_{\mathbf{p}_z}\left[\xi_{z,i}^2\right]^2 = \frac{16\gamma^4}{\alpha_{z,i}^4}\mathbb{E}\left[(\mathbf{m}_i - \mathbb{E}[\mathbf{m}_i])^2 \mathbb{1}\{\mathbf{m}_i > 10\mathbb{E}[\mathbf{m}_i]\}\right]^2$$

$$\leq \frac{4k^2}{m^2}\mathbb{E}\left[(\mathbf{m}_i - \mathbb{E}[\mathbf{m}_i])^4\right]\Pr\left(\mathbf{m}_i > 10\mathbb{E}[\mathbf{m}_i]\right)$$

$$\leq 64\Pr\left(\mathbf{m}_i > 10\mathbb{E}[\mathbf{m}_i]\right)$$

$$\leq 64e^{-4\mathbb{E}[\mathbf{m}_i]} \leq e^{-\frac{4m}{k}}$$

using that $\mathbb{E}\left[(N - \lambda)^4\right] = \lambda + 3\lambda^2$ for $N \sim \mathrm{Poi}(\lambda)$, and that $1 \leq \frac{m}{k} \leq \mathbb{E}[\mathbf{m}_i] \leq \frac{2m}{k}$. $\qquad\square$

For any fixed $z$ and $i$, in view of bounding the RHS of (16),

$$\sum_{i=1}^k \sum_{y \in \mathcal{Y}} \frac{\mathbb{E}_{\mathbf{p}_z}[\phi_{z,i}(X)W(y \mid X)]^2}{\mathbb{E}_{\mathbf{p}_z}[W(y \mid X)]}$$

$$\leq 3\sum_{i=1}^k \sum_{y \in \mathcal{Y}} \frac{\mathbb{E}_{\mathbf{p}_z}[\zeta_{z,i}(X)W(y \mid X)]^2}{\mathbb{E}_{\mathbf{p}_z}[W(y \mid X)]} + 3\sum_{i=1}^k \sum_{y \in \mathcal{Y}} \frac{\mathbb{E}_{\mathbf{p}_z}[\xi_{z,i}(X)W(y \mid X)]^2}{\mathbb{E}_{\mathbf{p}_z}[W(y \mid X)]}$$

$$+ 3\sum_{i=1}^k \sum_{y \in \mathcal{Y}} \frac{\mathbb{E}_{\mathbf{p}_z}[\psi_{z,i}(X)W(y \mid X)]^2}{\mathbb{E}_{\mathbf{p}_z}[W(y \mid X)]}. \tag{26}$$

We can now use our analysis above to handle those three terms. By subgaussianity, we have

$$\sum_{i=1}^k \sum_{y \in \mathcal{Y}} \frac{\mathbb{E}_{\mathbf{p}_z}[\zeta_{z,i}(X)W(y \mid X)]^2}{\mathbb{E}_{\mathbf{p}_z}[W(y \mid X)]} \leq O(\ell).$$

By a similar argument as in [4], by Cauchy–Schwarz we have

$$\sum_{i=1}^k \sum_{y \in \mathcal{Y}} \frac{\mathbb{E}_{\mathbf{p}_z}[\xi_{z,i}(X)W(y \mid X)]^2}{\mathbb{E}_{\mathbf{p}_z}[W(y \mid X)]} \leq \sum_{i=1}^k \sum_{y \in \mathcal{Y}} \mathbb{E}_{\mathbf{p}_z}\left[\xi_{z,i}^2(X)W(y \mid X)\right] = \sum_{i=1}^k \mathbb{E}_{\mathbf{p}_z}\left[\xi_{z,i}^2\right] \leq 8ke^{-2\frac{m}{k}},$$

and in the same way

$$\sum_{i=1}^k \sum_{y \in \mathcal{Y}} \frac{\mathbb{E}_{\mathbf{p}_z}[\psi_{z,i}(X)W(y \mid X)]^2}{\mathbb{E}_{\mathbf{p}_z}[W(y \mid X)]} \leq \sum_{i=1}^k \mathbb{E}_{\mathbf{p}_z}\left[\psi_{z,i}^2\right] \lesssim k\left(\frac{m}{k} + 1\right)\gamma^2 \leq 2m\gamma^2$$

Thus, we can further upper bound (26) as

$$\sum_{i=1}^k \sum_{y \in \mathcal{Y}} \frac{\mathbb{E}_{\mathbf{p}_z}[\phi_{z,i}(X)W(y \mid X)]^2}{\mathbb{E}_{\mathbf{p}_z}[W(y \mid X)]} \lesssim \ell + m\gamma^2 + ke^{-2\frac{m}{k}} \lesssim \ell + m\gamma^2, \tag{27}$$

the last bound using that $m \geq k \log k$.

Combining (27) with Theorem C.2 and Lemma C.5 and recalling that $\alpha^2 = O\left(m\gamma^2/k\right)$, we get $n\frac{m\gamma^2}{k}(\ell + m\gamma^2)) = \Omega(k)$, which implies

$$\gamma = \Omega\left(\min\left\{\sqrt{\frac{k^2}{mn\ell}}, \sqrt{\frac{k}{m\sqrt{n}}}\right\}\right) = \Omega\left(\sqrt{\frac{k^2}{mn\ell}}\right) \tag{28}$$

the last equality using our assumption that $n > (k/\ell)^2$. $\qquad\square$

## C.5 The lower bound for $k/2^\ell \leq m < k \log k$ and $n > (k/\ell)^2$

The last part missing is the proof of the lower bound for $k/2^\ell \leq m < k \log k$, which we give now. Unlike the previous two, this does not require any particular argument, but instead follows directly from the case $m \geq k \log k$:

**Theorem C.10.** *For all $m, n, \ell \geq 1$ such that $k/2^\ell \leq m < k \log k$ and $n > (k/\ell)^2$, the minimax rate in the multinomial setting satisfies*

$$\mathcal{R}(\ell, k, n, m) = \Omega\left(\sqrt{\frac{k}{mn}} \vee \sqrt{\frac{k}{n\ell \log k}}\right).$$

*Proof.* Again, the first term is just the standard centralized setting lower bound, and thus we can focus on establishing the second. The lower bound from Theorem C.7 implies the same lower bound for the multinomial setting, by Lemma C.1. Now, assuming this lower bound, we observe that the minimax rate is monotone nonincreasing in $m$ (all other parameters being fixed), and thus the bound obtained for $m_0 = k \log k$ does apply when $k/2^\ell \leq m < k \log k$. This leads to

$$\mathcal{R}(\ell, k, n, m) = \Omega\left(\sqrt{\frac{k^2}{m_0 n\ell}}\right) = \Omega\left(\sqrt{\frac{k}{n\ell \log k}}\right),$$

as long as $n > (k/\ell)^2$ (so that Theorem C.7 applies to $m_0, n, \ell, k$). This concludes the proof. $\square$