# OpenReview forum: "Distributed Estimation with Multiple Samples per User: Sharp Rates and Phase Transition"
_NeurIPS.cc/2021/Conference — NeurIPS 2021 Poster_

### Official Review · Reviewer_RoPp · 2021-07-13

**Rating:** 7
**Confidence:** 2

**Summary:**

The paper introduces new algorithms for low bandwidth distributed estimation problems which could be useful in different applications including distribution estimation via multiple sensor readings. The assumption that each node or user can have multiple samples makes this paper interesting and makes it different from the previous works. Another important assumption that makes the paper interesting is the low bandwidth restriction. The authors prove phase transition properties as the bandwidth increases.

**Ethical Concerns:**

No concern

**Limitations And Societal Impact:**

This paper seems to be among the first works in this area. So it studies a relatively easy case and proposes algorithms for this case. It is therefore limited to such cases. Future works could improve upon the results in this paper.

**Main Review:**

The big O notation is not carefully used throughout the paper. In some occasions the authors need to clarify about the parameter that goes to infinity in the Big O.

There is no comparison with already existing bounds. It is recommended that the authors include the existing crude bounds and then compare it with their proposed bounds to show the amount of improvement in the computed bounds.

**Time Spent Reviewing:**

5

---

> ### Author Response · Authors · 2021-08-08
> **Response to reviewer RoPp**
>
> Thanks for the feedback. We will make the presentation clearer in the updated version, and clarify our use of asymptotic notation. In terms of existing bounds, the setting where each user has $m > 1$ samples is fairly new and, to the best of our knowledge, there are no nontrivial bounds in the literature, even crude ones. The most closely related works provide bounds in the cases when $m = 1$ (one sample per user) or $\ell = \infty$ (centralized setting), which are listed in Equation (2) and (3).

---

### Official Review · Reviewer_VJXc · 2021-07-14

**Rating:** 6
**Confidence:** 3

**Summary:**

This work considers distributed density estimation. The problem setting is as follows: n users each have a dataset of m independent observations from an unknown discrete probability distribution and a central server aims to estimate the unknown distribution. When each user is subject to a bandwidth constraint such that each user can only send \ell bits to the center, the authors seek to understand tradeoffs between communication, sample size, and estimation error in the problem. In particular, for an unknown $k$-ary distribution p, this works characterizes minimax rates of estimating p in L1 distance.


**Limitations And Societal Impact:**

Yes

**Main Review:**

I think the problem studied studied in the paper, namely that of distributed density estimation, is interesting and timely. I believe the work provides significant generalizations beyond what exists currently in the literature (existing literature usually uses either users having m=1 sample or one able to send \ell=1 bit of info), and this requires some novel proof ideas. For what it's worth, I also think the work is nice because it pulls high level ideas from many different disciplines like information theory, statistics, and applied probability. I list a few comments below followed by some typos I found.

Some comments on the work:

One comment is that at some points in the paper, I thought the writing wasn't very accessible to non-experts. Essentially, if you don't already understand the problem set-up and some standard coding schemes for such problems, you will likely have difficulty understanding the first section of the paper. For example, in introducing the problem setting, it is not clear what you mean by the channel. Is your setting that a central server receives a noisy version of each user's message y_i? This is one standard usage of 'channel' in communication and statistics, but apparently not what is meant here. In particular, you don't tell use what is meant by x in your problem setting in the definition of the channel between lines 36-37. It may be useful to say here something like the channel represents the decision making process of the user: it states the probability distribution relating each possible user dataset to each possible user message. As another example, you aim to motivate your algorithm with a simple estimation problem related to coin tosses, but then when explaining how to solve this estimation problem you end up just saying it's technical and "we use a scheme similar to the Gray coding based scheme proposed for Gaussian mean estimation in [7], but with modification." So unless the reader is familiar with the Gray coding scheme, this doesn't do much in the way of motivation. A way to actually motivate the suggested scheme might be to discuss why a naive solution of having each user send 0 if they see more heads than tails and 1 otherwise is ineffective if the coin is strongly biased. This would at least give the reader a feel for what the technical challenges are and some intuition for the two-pronged approach suggested in what follows.

Some typos/comments:
-- You use the notation \textbf{p} in line 23, but it isn't defined until later.
-- What is $\mathcal{W}$ in line 35?
-- Should the O() rate between lines 168-169 use sum_x p_x(1-p_x) in the first term in line with the result between lines 164-165?
-- On line 179 "... is defined by a a collection..."
-- It may be worth mentioning that your proof of Lemma 3.1 uses a different (but equivalent) definition of total variation than is given in the problem setting and notation section. I think the second inequality in the proof is actually an equality.


**Time Spent Reviewing:**

2

---

> ### Author Response · Authors · 2021-08-08
> **Response to reviewer VJXc**
>
> Thanks a lot for the constructive feedback. We will make the presentation more accessible based on your suggestions, specifically in clarifying our use of channel (randomized mapping from input to output, which the user gets to select under the constraints they must satisfy -- your example corresponds to the specific case of a noisy communication channel, where the user has little control over the choice, but the channel is a randomized mapping which flips or erases bits of the input), and introducing the Gray coding scheme in a more intuitive way by discussing what goes wrong with a more naive and natural approach.
>
> As you point out, the second inequality in the proof of Lemma 3.1 should be an equality. We will fix the typos pointed out in the review.

---

> ### Comment · Reviewer_VJXc · 2021-08-22
> **Response to Authors**
>
> Thanks to the authors for their thoughtful responses to my and the other referee's reviews of the work.  My review remains unchanged.

---

### Official Review · Reviewer_RBWm · 2021-07-17

**Rating:** 6
**Confidence:** 3

**Summary:**

The paper studies the problem of distributed learning of a discrete distribution.  Formally, there are $n$ participants each of whom receives $m$ samples from a discrete distribution $\mathcal{D}$  on $[k]$.  The participants, after seeing their own samples, each send at most $l$ bits of information to a centralized server.  The goal is for the centralized server to output a distribution $\mathcal{D'}$ that is close to $\mathcal{D}$ in TV distance.

The main results of the paper are matching upper and lower bounds in various regimes of $n,m,k, l$, generalizing previous works that had only studied the case of $m = 1$.  Furthermore, the upper bounds are obtained using non-interactive algorithms with no shared randomness while the lower bound holds even against sequential algorithms that use shared randomness.  The main takeaway is a supposed phase transition when $m \sim k/2^l$ where the "value" of additional bits goes from exponential when $m < k/ 2^l$ to polynomial when $m > k/2^l$.

**Main Review:**

The proposed problem is a nice and natural generalization of the $m = 1$ case that had been studied previously.  Overall, the presentation is reasonable.

I do have a few questions about the main results.  Theorem 1.1 solves the case when $m < k/2^l$.  However, I have some concerns about the significance of Theorems 1.2 and 1.3 (especially Theorem 1.3) and whether they actually imply the phase transition that is claimed.

1. In both statements (Theorem 1.2 and Theorem 1.3) should the $\sqrt{\frac{k^2}{mn}}$ be $\sqrt{\frac{k}{mn}}$ instead?  A lower bound of  $\sqrt{\frac{k^2}{mn}}$ seems to be false in the regime where $n = 1$ and one player may just communicate the entire distribution to the central server.  Does the upper bound proof also only need  $\sqrt{\frac{k}{mn}}$ and not $\sqrt{\frac{k^2}{mn}}$  (based on skimming the supplement this seems to be the case but I did not verify all of the details carefully)?

2. My main concern is the following:  if I want to estimate the distribution to some constant accuracy  (say $0.1$) then the lower bound in Theorem 1.3 only kicks in when $l \geq k$.  This is because by assumption $n> \frac{k^2}{l^2} \rightarrow nl > \frac{k^2}{l}$ so plugging this into the expression for the lower bound gives something less than $1/(\log k)$ unless $l \geq k$.  However, $l \geq k$ is well outside the range of the claimed phase transition at $m \sim k/2^l$  (which only makes sense for $l \leq \log k$).  Thus, it does not seem that we can apply Theorem 1.3 to prove that there is indeed a phase transition.

After Author Response: Thanks for clarifications, after reading the response I do believe that the contributions are significant enough and I have raised my score accordingly.  Still, I think the authors need to be more clear and explicit about what regimes their results do and do not cover.  In particular the paper is a bit misleading in that it seems to claim a "full understanding" while the results only cover some of the important regimes.

**Time Spent Reviewing:**

2

---

> ### Author Response · Authors · 2021-08-08
> **Response to reviewer RBWm**
>
> We thank the reviewer for the comments. Indeed, point (1) is a (very unfortunate) typo on our side: the first term in both theorems should be the centralized complexity, $\sqrt{k/mn}$ (not $k^2$). All our proofs and following statements, both upper and lower bounds, establish the correct sample complexity with a $\sqrt{k/mn}$ term, and the error only appears in the two theorem statements.
>
> Point 2 is quite subtle, and we will clarify it in the final version of the paper. As pointed out, the phase transition we claim is not shown in all parameter regimes. In particular, our theorem does not establish it for the constant-error regime. Although we suspect it does exist, and the condition $n > (k/\ell)^2$ is an artifact of our proof. To see this, consider the case when $m = \infty$. Each user is able to learn the distribution themselve locally to arbitrary accuracy. However, to achieve a constant (say 0.1) accuracy at the server, by a packing argument, they still need to send at least $\Theta(k)$ bit in total to communicate a distribution with the desired accuracy to the central server, which shows that the $k^2/2^\ell$ rate cannot always hold.
>
> However, Theorem 1.3 does prove a phase transition in various parameter regimes, including the vanishing error one.
> Phrased in terms of sample complexity, for instance, we can rephrase this as saying that for eps “small enough,” the sample complexity goes from
>            $\frac{k^2}{m 2^\ell \varepsilon^2}$
> to
>            $\frac{k}{\ell \varepsilon^2}\log(2k/m)$
> With the transition at $m\approx k/2^\ell$; where “small enough” can be taken as $\varepsilon \leq O(1/\sqrt{k})$.
>
> As a specific example, consider the case $n = k^2$, where the condition in Theorem 1.3 is satisfied for all $\ell$. In this case, when $m < k/2^{\ell}$, the rate we obtain is $\Theta(\frac{1}{ m 2^\ell})$ according to Theorem 1.1. When $m > k\log k$, we can show the risk is $\Theta(\frac{1}{m \ell})$ according to Theorem 1.2 and 1.3. Hence there is indeed a phase transition in-between. Moreover, note that when $k/2^{\ell} < m < k\log k$, we show the rate is $\tilde{\Theta}(1/ (k \ell))$, only depending on $m$ through logarithmic terms, which can be viewed as another phase transition in terms of $m$.
>
> Finally we would like to emphasize that extending the distribution estimation problem to multiple samples per user is far from obvious, and involves overcoming quite a few obstacles (as hinted to by the different parameter regimes in the rates, that none of the obvious “natural” approaches would result in). We are, to the best of our knowledge, the first to study the problem, and the algorithm we propose is very different from the approach followed in prior work on discrete distribution estimation. The phase transition results we show, albeit for limited parameter ranges, uncover an interesting phenomenon and hint towards why multiple-sample problems are significantly more involved than the single-sample version.

---

### Official Review · Reviewer_EXUY · 2021-07-18

**Rating:** 7
**Confidence:** 4

**Summary:**

This paper is about distributed density estimation of an unknown probability distribution $p$, where the samples from that distribution are held by different users. In the model considered, there is a central server that collects messages from the users who hold the samples from $p$. Apart from the number of users $n$ and the domain size $k$, the problem is parametrized by the number of samples $m$ that each user holds and the maximum message size of $\ell$ bits. The goal is to minimize the total communication complexity (i.e the number of bits sent to the server). Interestingly, the authors show a phase transition of the error of the estimation with regard to $m$ happening at $m=k/2^\ell$. Specifically, it is shown that the error scales with $\frac{1}{\sqrt{\ 2^\ell}}$ for $m<k/2^\ell$ and with $\frac{1}{\sqrt{\ell}}$ otherwise. The existence of this phase transition is supported by both upper and lower bounds, although they are not shown for the entire parameter regime. However, both bounds are shown for sufficiently large $n$ and they are also tight up to logarithmic factors. The algorithm uses the estimation of a cointoss ($k=2$) bernoulli distribution as a primitive and recursively estimates the mass of certain parts of the domain in this way.




**Limitations And Societal Impact:**

The authors as well as myself do not see any potential negative societal impact from this work.

**Main Review:**

The paper makes some progress on the very fundamental problem density estimation of a probability distribution in the distributed setting. The main contribution is the extension to the case where each user holds multiple samples (denoted by $m$) from the distribution, rather than just one sample as in prior works. This also revealed an interesting phase transition of the error of the estimation around $m=k/2^\ell$, where $\ell$ is the message bound. The authors also provide lower bounds showing that the phase transition is inherent. On the negative side, some of the bounds need artificial conditions to work and although tightness up to log factors is shown for sufficiently large $n$, there are still parameter regimes where the bounds do not apply.


Minor Comments:
Line 4: It might be useful to introduce the notation k, for the domain size here along with the other parameters.
Lines 32 and 36: It might be confusing that the same variable (x) is used both as an integer and as a vector.
Line 35: $\mathcal{W}$->$\mathcal{W}_\ell$, as it is used later in equation (1)
Line 88: can “be” shown
Line 112: $\ell$ “bits” of communication


**Time Spent Reviewing:**

3

---

> ### Author Response · Authors · 2021-08-08
> **Author response to Reviewer EXUY**
>
> Thanks for the positive reviews. We will make the changes suggested by the reviewer to introduce the notation and fix the typos they pointed out.

---

### Decision · Program_Chairs · 2021-09-27

**Decision:**

Accept (Poster)

**Comment:**

Overall the reviewers liked the matching upper and lower bounds in a wide range of regimes for a quite natural distributed problem, generalizing in a significant way previous work that only held for one sample per machine. The reviewers also found the phase transition to be interesting, as well as the techniques drawn from different areas. There were some specific presentation comments that we encourage the authors to take into account.